# *Metis*: A Foundation Speech Generation Model with Masked Generative Pre-training

**Yuancheng Wang, Jiachen Zheng, Junan Zhang, Xueyao Zhang, Huan Liao, Zhizheng Wu**
The Chinese University of Hong Kong, Shenzhen
yuanchengwang@link.cuhk.edu.cn, wuzhizheng@cuhk.edu.cn

## Abstract

We introduce *Metis*, a foundation model for unified speech generation. Unlike previous task-specific or multi-task models, Metis follows a pre-training and fine-tuning paradigm. It is pre-trained on large-scale unlabeled speech data using masked generative modeling and then fine-tuned to adapt to diverse speech generation tasks. Specifically, 1) Metis utilizes two discrete speech representations: SSL tokens derived from speech self-supervised learning (SSL) features, and acoustic tokens directly quantized from waveforms. 2) Metis performs masked generative pre-training on SSL tokens, utilizing 300K hours of diverse speech data, without any additional condition. 3) Through fine-tuning with task-specific conditions, Metis achieves efficient adaptation to various speech generation tasks while supporting multimodal input, even when using limited data and trainable parameters. Experiments demonstrate that Metis can serve as a foundation model for unified speech generation: Metis outperforms state-of-the-art task-specific or multi-task systems across five speech generation tasks, including zero-shot text-to-speech, voice conversion, target speaker extraction, speech enhancement, and lip-to-speech, even with fewer than 20M trainable parameters or 300 times less training data. Audio samples are are available at https://metis-demo.github.io/. We release the code and model checkpoints at https://github.com/open-mmlab/Amphion.

## 1 Introduction

Advancing a unified framework capable of addressing diverse tasks is a central research objective within the domain of artificial intelligence. In natural language processing [1, 2, 3] and computer vision [4, 5, 6], foundation models leveraging large-scale self-supervised pre-training have demonstrated remarkable adaptability across a wide spectrum of downstream tasks. However, in the domain of speech generation, this potential remains underexplored. A unified speech can integrate various speech generation technologies, such as text-to-speech [7, 8, 9, 10, 11, 12], voice conversion [13, 14, 15, 16], and speech enhancement [17, 18, 19]. This integration reduces redundant development, facilitates broader applications across diverse domains, and enhances the efficiency of human-machine interaction.

Previous speech generation models are mostly expert models that require extensive task-specific designs [7, 20, 21, 22, 23, 24, 25]. UniAudio [26] and SpeechX [27] are pioneering works that attempt to use autoregressive language models for multiple speech generation tasks. However, *they require a large amount of paired training data for each task and face challenges in extending to new tasks based on pre-trained models*. In addition, the autoregressive approach leads to suboptimal results in certain tasks and is relatively inefficient.

In this paper, we address the research question of ***how to design a unified speech generation framework that leverages large-scale unlabeled speech data for pre-training and efficiently adapts to diverse speech generation tasks through fine-tuning.*** Inspired by previous two-stage speech

39th Conference on Neural Information Processing Systems (NeurIPS 2025).

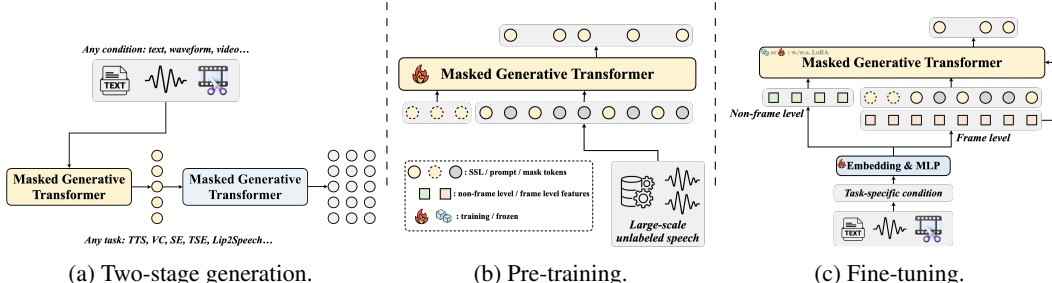

|  (a) Two-stage generation. | (b) Pre-training. | (c) Fine-tuning. |

Figure 1: An illustration of Metis. (a) provides an overview of the two-stage speech generation framework, which consists of task-specific (yellow block) and task-independent (light blue block) processes. In this work, we focus on developing a pre-training model for the first stage, as illustrated in (b). (c) demonstrates the fine-tuning process, where the pre-trained model is adapted to specific tasks.

generation models [28, 29, 30, 31, 32], we revisit the speech generation process and observe a common structure underlying most tasks: generating intermediate representations from task-specific conditions and synthesizing acoustic representations from these intermediates. The intermediate representations are typically discrete units quantized from self-supervised speech features [33, 34, 35, 36, 37] that encode semantic and prosodic information, while acoustic representations are easily used to reconstruct high-quality waveforms.

Building on this observation, we propose a unified framework that modularizes speech generation tasks as a two-step process, as illustrated in Figure 1a: 1) **Task-Specific Process**: Generating SSL tokens conditioned on task-specific conditions (e.g., text for text-to-speech, noisy speech for speech enhancement, or visual features for lip-to-speech). 2) **Task-Independent Process**: Generating acoustic representations from the SSL tokens. This formulation allows diverse tasks to be addressed by varying input conditions while maintaining a consistent generation mechanism. Previous studies [12, 28, 30, 31, 32, 38, 39, 40] have established that the second stage of acoustic refinement can be achieved effectively using fully self-supervised diffusion models [31, 32, 38, 39, 40] or language models [12, 28, 30]. Therefore, *our objective is simplified to designing a generative pre-training mechanism for the first stage, enabling the pre-trained model to effectively generalize across various speech generation tasks*. In this work, we adopt masked generative models [41] for unconditional pre-training. Masked generative models are trained using a straightforward masked token prediction approach and employ an iterative sampling strategy to generate outputs from a fully masked input. We provide more details about masked generative models in Section 3.1. Once pre-trained, the model can be fine-tuned with limited data and parameters by incorporating task-specific conditions as additional inputs to adapt to various speech generation tasks.

Based on the above discussion, we propose *Metis*, a foundation model for unified speech generation. Specifically, Metis has the following key features:

**Masked Generative Pre-Training:** Metis performs masked generative pre-training on SSL tokens using large-scale unlabeled speech data without any task-specific condition, as illustrated in Figure 1b. This pre-training phase establishes a strong foundation, allowing Metis to efficiently adapt to various downstream tasks through fine-tuning with minimal task-specific data.

**Efficient Adaption to Various Speech Generation Tasks:** Metis can be efficiently adapted to a wide range of speech generation tasks, including zero-shot text-to-speech, voice conversion, speech enhancement, target speaker extraction, and lip-to-speech, as illustrated in Figure 1c. The fine-tuned models achieve state-of-the-art results, even when using fewer than 20M trainable parameters or significantly less training data.

**Support for Multimodal Conditional Inputs:** Metis supports multimodal conditional inputs during the fine-tuning phase, including text, audio, and video. This capability solidifies Metis as a foundation model for supporting various speech generation tasks. We also explore multi-task fine-tuning, showing that the pre-trained model can be efficiently adapted into a powerful multi-task model with minimal modification, enabling novel applications such as text-guided target speaker extraction with task combinations.

## 2 Related Work

**Masked Generative Models for Speech**   Masked generative models (MGMs) are a family of generative models that typically employ non-autoregressive transformers [42]. These models have achieved significant success, demonstrating performance comparable to or even surpassing autoregressive and diffusion models in image [41, 43, 44] and video [45, 46] generation, while offering a better balance between quality and speed. In the speech domain, SoundStorm [30] uses the semantic tokens from AudioLM [28] and employs MGMs to generate acoustic tokens from a neural audio codec [47], enabling applications like TTS and voice conversion. NaturalSpeech 3 [10] adopts MGMs to generate disentangled speech tokens. MaskGCT [12] further leverages MGMs for zero-shot generation, eliminating the need for explicit text-speech alignment or phone-level duration prediction in non-autoregressive TTS models. MaskSR [48] applies MGMs to speech enhancement tasks. In this work, we propose a unified speech generation framework based on MGMs.

**Unified Speech Generation**   Developing a unified framework capable of handling various tasks is a key research objective in artificial intelligence. In the field of speech generation, UniAudio [26] employs an LLM for next-token prediction to generate multiple types of audio. Similarly, SpeechX [27] leverages an LLM for unified zero-shot tasks such as TTS, noise suppression, and target speaker extraction. Both models achieve this by concatenating the condition and target speech tokens, followed by AR modeling. However, these models require large amounts of paired training data for each task, failing to leverage the vast amount of unlabeled speech data effectively. VoiceBox [11] employs flow matching to unify tasks such as zero-shot TTS, speech editing, and speech enhancement. However, it has notable limitations, such as requiring text and clean speech as references for speech enhancement and relying on phone durations during training for zero-shot TTS. Its successor, AudioBox [49], extends VoiceBox to unified audio generation with natural language prompt control. Our work is partly inspired by SpeechFlow [50], which uses flow matching [51] to learn infilling during pre-training and fine-tunes with task-specific conditions for various speech generation tasks, such as zero-shot TTS and speech separation. However, it is limited to frame-level conditions, such as requiring a frame-level phoneme sequence for TTS. Additionally, predicting mel-spectrograms directly during pre-training may be suboptimal due to the need to predict extensive acoustic details.

Speech discrete representation is also highly relevant to our work, and we have included this part in Appendix F.

## 3 Method

### 3.1 Background: Masked Generative Models

In this section, we provide a brief introduction to masked generative models [12, 41, 46]. Consider a discrete sequence $x = [y_1, y_2, \ldots, y_n]$, where $n$ denotes the length of the sequence. We define $x_t = x \odot m_t$ as the operation of masking a subset of tokens in $x$ using the corresponding binary mask $m_t = [m_{t,1}, m_{t,2}, \ldots, m_{t,n}]$. Specifically, this operation involves replacing $x_i$ with a special [MASK] token if $m_{t,i} = 1$, and otherwise leaving $x_i$ unmasked if $m_{t,i} = 0$. Here, each $m_{t,i}$ is independently and identically distributed according to a Bernoulli distribution with parameter $\gamma(t)$, where $\gamma(t) \in (0, 1]$ represents a mask schedule function (for example, $\gamma(t) = \sin(\frac{\pi t}{2T}), t \in (0, T)$). We denote $x = x_0$. The masked generative models are trained to predict the complete sequence (masked tokens) based on the observed tokens (unmasked tokens) and the condition $c$, which can be modeled as $p_\theta(x_0 \mid x_t, c)$, and the model parameters $\theta$ are trained to optimize the sum of the marginal cross-entropy for each unobserved token:

$$\mathcal{L}_{\text{mask}} = -\mathbb{E}_{x,t,m_t} \sum_{i=1}^{n} m_{t,i} \cdot \log p_\theta(y_i \mid x_t, c) \tag{1}$$

Note that $c$ may be empty, for example, during the unconditional pre-training stage of our model. At the inference stage, masked generative models generate tokens in parallel through iterative decoding. The process begins with a fully masked sequence $x_T$. Assuming the total number of decoding steps is $S$, for each step $j$ from 1 to $S$, we first sample $\hat{x}_0$ from $p_\theta(x_0 \mid x_{T-(j-1)\cdot\frac{T}{S}}, c)$. Then we sample $\lfloor n \cdot \gamma(T - j \cdot \frac{T}{S}) \rfloor$ tokens based on the confidence score to remask, resulting in $x_{T-j\cdot\frac{T}{S}}$, where $n$ is the sequence length of $x$. The confidence score for $\hat{y}_i$ in $\hat{x}_0$ is assigned to $p_\theta(y_i \mid x_{T-(j-1)\cdot\frac{T}{S}}, c)$

if $y_{T-(j-1)\cdot\frac{T}{S},i}$ is a [MASK] token; otherwise, we set the confidence score of $\hat{y}_i$ to 1, indicating that tokens already unmasked in $\boldsymbol{x}_{T-(j-1)\cdot\frac{T}{S}}$ will not be remasked. In particular, we choose $\lfloor n\cdot\gamma(T-j\cdot\frac{T}{S})\rfloor$ tokens with the lowest confidence scores to be masked. Note that the method for calculating the confidence score is not unique. For example, Lezama et al. [52] introduces Token-Critic, training a critic model to compute confidence scores, aiding the sampling process. Additionally, Xie et al. [44], Lezama et al. [52] suggest that masked generative modeling can be seen as a simplified version of discrete diffusion.

## 3.2 Overview of Metis

Figure 1 provides an overview of our system, which adopts a two-stage speech generation process for unified speech generation tasks. We first briefly introduce the distinct speech discrete representations used in the two stages in Section 3.3. Then, we present the pre-training (Section 3.4) and fine-tuning (Section 3.5) strategies for the first stage, which form the core of our work. Finally, we provide a brief overview of a unified acoustic decoder for all speech generation tasks in Section 3.6.

## 3.3 Discrete Representations for Two-Stage Generation

Metis employs two discrete speech representations for the two-stage generation, as illustrated in Figure 2. 1) **SSL tokens**: Derived from SSL features of large-scale speech self-supervised learning models [33, 34, 36, 37], SSL tokens encapsulate both semantic and prosodic information, making them well-suited for conditional generation. To minimize information loss, we employ a vector quantization (VQ) model [53, 54] to quantize SSL features into discrete tokens, following Wang et al. [12]. 2) **Acoustic tokens**: Directly obtained from the waveform via vector quantization. The goal is to preserve all the information from the speech to reconstruct a high-quality waveform. We show more details in Appendix A.

## 3.4 Masked Generative Pre-training with SSL Tokens

Based on the previous discussion, most speech generation tasks can be generalized into two stages: *conditions to SSL tokens* and *SSL tokens to acoustic tokens*. The primary distinction among tasks lies in the nature of the conditions. To address this, we propose a unified pre-trained model for the first stage, which can be adapted to various tasks.

We use an unconditional masked generative model on SSL tokens for pre-training. Specifically, we randomly mask tokens in the SSL token sequence $\boldsymbol{x}^{ssl}$ using the strategy outlined in Section 3.1 and predict the masked tokens. We introduce a prompt sequence with the probability $p$ to further enhance the in-context learning ability of the model. With this probability, a prefix sequence $\boldsymbol{x}^{ssl}_{prompt}$ from the SSL token sequence is used as a prompt and remains unmasked. This mechanism enables the model to leverage prompt information, thereby improving its adaptability to downstream tasks requiring prompts, such as zero-shot TTS and target speaker extraction. The pre-training objective is to model $p_\theta(\boldsymbol{x}^{ssl}_0 \mid \boldsymbol{x}^{ssl}_t, \boldsymbol{x}^{ssl}_{prompt})$.

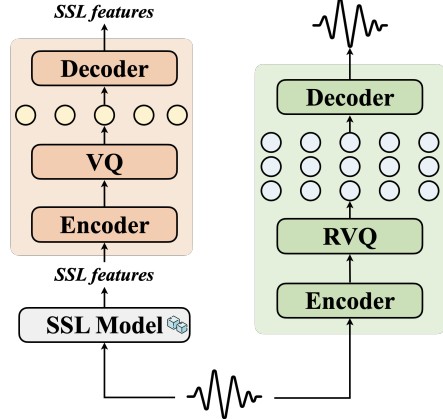

Figure 2: Two discrete speech representations for the two-stage speech generation: SSL tokens (left) and acoustic tokens (right).

Our design is motivated by the observation that models trained on extensive data can recover masked SSL tokens from prompts and unmasked tokens, even without task-specific conditions, as shown in other domains [2, 4]. Empirically, the pre-trained model can generate speech that mimics the prosodic style and timbre of a prompt. However, the generated speech often lacks intelligibility, producing random word concatenations due to the absence of semantic guidance. This highlights the need for task-specific conditions, which can be efficiently incorporated through fine-tuning to adapt the model to various speech generation tasks.

### 3.5 Efficient Adaptation to Various Generation Tasks

Now, we describe how to efficiently adapt the pre-trained model to various speech generation tasks. We first categorize the conditions for different speech generation tasks into two types: non-frame-level conditions and frame-level conditions. For the former, such as TTS, the condition is a phoneme sequence or a text token sequence. In this case, the model needs to implicitly learn the alignment between the condition and the SSL token sequence. For the latter, such as voice conversion or speech enhancement, the conditions (e.g., the source speech voice conversion or the noisy speech for speech enhancement) can be aligned with the target SSL token sequence at the frame level. Based on these distinctions, during fine-tuning, for non-frame-level conditions, we simply concatenate the condition with the input sequence along the time dimension. For frame-level conditions, we apply a simple interpolation to align the condition with the input sequence in the time dimension and then pass it through an MLP-based adapter before adding it to the input. Then, the fine-tuned model is trained to learn $p_\theta(x_0^{ssl} \mid x_t^{ssl}, x_{prompt}^{ssl}, c)$, where $c$ is the task-specific condition. Our experiments demonstrate that the pre-trained model can efficiently adapt to various tasks and achieve remarkable results with minimal data. Additionally, we explored the use of Low-Rank Adaptation (LoRA) [55], which allows fine-tuning with a small number of trainable parameters. More details are provided in Section 4.

### 3.6 Masked Generative Acoustic Decoder

We train an SSL-to-acoustic model based on masked generative modeling, which serves as a unified acoustic decoder for all speech generation tasks. The model is trained to recover masked tokens from a masked acoustic token sequence $x_t^a$, conditioned on SSL tokens $x_{ssl}$ and prompt acoustic tokens $x_{prompt}^a$. This can be formulated as $p_\theta(x_0^a \mid x_t^a, x_{prompt}^a, x_{ssl})$. During training, we randomly select a layer for masking in the multi-layered acoustic tokens, while the lower-layer tokens remain unmasked and serve as conditional inputs to the model, following [30]. During inference, we generate acoustic tokens layer by layer. Finally, we generate the speech by first predicting SSL tokens and then generating acoustic tokens.

## 4 Experiments and Results

### 4.1 Setup

**Model Architecture**  We adopt the same model architecture as described in Wang et al. [12], except removing the text embedding during the pre-training phase. The model follows the standard Llama-style architecture [42, 56], but replaces causal attention with bidirectional attention.

**Dataset**  We use a dataset consisting of 300K hours of speech for pre-training, including 100K hours from the Emilia [57] dataset and an additional 200K hours self-collected through the Emilia pipeline, to train our pre-trained model. The dataset contains a diverse range of in-the-wild multilingual speech data, including 87K hours in Chinese, 185K hours in English, 7K hours in German, 8K hours in French, 2.5K hours in Japanese, and 7.5K hours in Korean. The dataset used for fine-tuning is sampled from the pre-trained dataset.

**Training**  We pre-train our model on 8 GPUs for a total of 1200K steps. We use the AdamW [58] optimizer with a learning rate of 1e-4 and 32K warmup steps. We employ a dynamic batch size, where each batch contains 10K tokens (200 seconds) per GPU. During training, we randomly select a prefix of the sequence as a prompt that is not masked with a probability $p = 0.8$. The length of the prompt is uniformly sampled from the range $[0\%, 40\%]$ of the total sequence length.

**Inference**  We follow the inference strategy outlined in Wang et al. [12], with task-specific step adjustments for the fine-tuned model to generate SSL tokens. Additionally, we use classifier-free guidance [59] for tasks that involve prompts.

**Evaluation Metrics**  We use multiple evaluation metrics to assess different aspects of the generated speech, including similarity (SIM), intelligibility (WER), and audio quality (DNSMOS, SIG, BAK, OVRL, NISQA). For the subjective metrics, quality mean opinion score (CMOS) and similarity mean

opinion score (SMOS) are used to evaluate speech quality and similarity. The results of the subjective evaluation are presented in Appendix D.3. Details about these metrics are provided in Appendix C.

## 4.2 Results on Different Speech Generation Tasks

We present the fine-tuning results of Metis on various speech generation tasks. These tasks include zero-shot TTS (without frame-level phoneme condition or duration prediction), voice conversion, speech enhancement, target speaker extraction, and the multimodal task of lip-to-speech.

### 4.2.1 Zero-Shot TTS

**Implementation Details**  For zero-shot TTS, we fine-tune the pre-trained model on three different datasets: 1K hours and 10K hours randomly sampled from the Emilia dataset, and the LibriTTS [62] dataset, which contains 0.58K hours of English speech data. We use two fine-tuning methods: full-scale fine-tuning and LoRA fine-tuning. For LoRA fine-tuning, we set the rank $r = 32$, resulting in only **32M trainable parameters**, including the newly added text embedding module. All fine-tuned models are trained using 4 GPUs for 200K steps.

**Evaluation and Baseline**  We evaluate our zero-shot TTS models using three test sets: 1) SeedTTS *test-en*, a test set introduced in Seed-TTS [31] comprising 1,000 samples extracted from English public corpora, including the Common Voice dataset [63]. 2) SeedTTS *test-zh*, a test set introduced in Seed-TTS consisting of 2,000 samples extracted from Chinese public corpora, including the DiDiSpeech dataset [64]. 3) LibriSpeech *test-clean* [65], a widely

Table 1: Results on zero-shot TTS task.

| Model | Training Data | WER(↓) | SIM(↑) | DNSMOS(↑) |
|---|---|---|---|---|
| *SeedTTS test-en* | | | | |
| Ground Truth | - | 2.14 | 0.73 | 3.53 |
| VALL-E [8] | 45K EN | 6.13 | 0.43 | 3.39 |
| VoiceCraft [60] | 9K EN | 7.55 | 0.47 | 3.37 |
| CosyVoice [32] | 170K Multi. | 4.08 | 0.64 | 3.64 |
| XTTS-v2 [61] | 27K Multi. | 3.25 | 0.46 | 3.45 |
| MaskGCT [12] | 100K Multi. | 2.47 | 0.72 | 3.51 |
| Metis-TTS LoRA 32 | 1K Multi. | 4.78 | 0.72 | 3.47 |
| Metis-TTS LoRA 32 | 10K Multi. | 4.55 | 0.72 | 3.45 |
| Metis-TTS LoRA 32 | 0.58K[1] EN. | 4.63 | 0.70 | 3.51 |
| Metis-TTS fine-tune | 0.58K[1] EN. | **3.04** | 0.68 | 3.47 |
| Metis-TTS fine-tune | 1K Multi. | 3.86 | 0.71 | 3.46 |
| Metis-TTS fine-tune | 10K Multi. | **2.28** | 0.72 | 3.47 |
| Metis-TTS w.o. pre-train | 10K Multi. | 4.91 | 0.69 | 3.42 |
| *SeedTTS test-zh* | | | | |
| Ground Truth | - | 1.25 | 0.75 | 3.51 |
| CosyVoice [32] | 170K Multi. | 4.09 | 0.75 | 3.71 |
| XTTS-v2 [61] | 27K Multi. | 2.88 | 0.63 | 3.44 |
| MaskGCT [12] | 100K Multi. | 2.18 | 0.77 | 3.58 |
| Metis-TTS LoRA 32 | 1K Multi. | 5.21 | 0.77 | 3.57 |
| Metis-TTS LoRA 32 | 10K Multi. | **4.48** | 0.77 | 3.55 |
| Metis-TTS fine-tune | 1K Multi. | 4.23 | 0.77 | 3.54 |
| Metis-TTS fine-tune | 10K Multi. | **2.30** | 0.77 | 3.55 |
| Metis-TTS w.o. pre-train | 10K Multi. | 4.98 | 0.73 | 3.51 |

[1] This version of the model is trained on LibriTTS [62].

used test set for TTS. We compare our models with several recent zero-shot TTS models, including AR-based models: VALL-E [8], VoiceCraft [60], XTTS-v2 [61], and CosyVoice [32]; NAR-based models: VoiceBox [11], and MaskGCT [12]. Specifically, MaskGCT can be seen as Metis without pre-training. For VoiceBox and NaturalSpeech 3, we only compare with them on LibriSpeech *test-clean* since they are not open-source. See more details about these baselines in Appendix B.1.

**Result**  The results are presented in Table 1, with additional results for LibriSpeech *test-clean* set provided in Appendix D.1. The table shows that Metis-TTS LoRA 32, fine-tuned with only 1K hours of data and 32M trainable parameters, achieves performance on metrics such as WER, SIM, and DNSMOS that is comparable to or exceeds that of some baselines trained on datasets 10 to 100 times larger. Furthermore, when scaled with full fine-tuning on 10K hours of data, Metis demonstrates an improved WER on SeedTTS *test-en*, outperforming MaskGCT, which was trained on 100K hours of data, while maintaining comparable WER performance on SeedTTS *test-zh*. When fine-tuned with only 580 hours of LibriTTS data, Metis achieves competitive WER and SIM performance, surpassing all baselines except MaskGCT on SeedTTS *test-en*. We also compare with models that are not pre-trained but trained for the same number of steps. The results show that the fine-tuned models converge faster and achieve better performance. In addition, all fine-tuned models exhibit excellent SIM and DNSMOS, indicating high speech similarity and quality.

### 4.2.2 Voice Conversion

**Implementation Details**  Previous voice conversion systems typically extract timbre-independent features from input signals using methods such as information bottlenecks [14, 66, 67], timbre perturbation [15, 68, 69], or specialized loss functions [10, 67]. However, these approaches often introduce complexity, risk the loss of semantic information, and may still inadvertently retain timbre-

related details [70, 71]. In this work, we use a simpler yet effective solution inspired by Anastassiou et al. [31], Neekhara et al. [72]. Specifically, we leverage a lightweight voice conversion model[1] [73] to perform real-time voice conversion on the target speech using a randomly sampled prompt speech, thereby achieving timbre perturbation. The perturbed speech features are then used as input to predict the target speech based on the prompt. Specifically, we employ the `w2v-bert-2.0` features of the perturbed speech as conditioning inputs and randomly extract a prefix of the target speech as the prompt. Our experiments demonstrate that with minimal data and training steps, our pre-trained model can effectively and rapidly adapt to the voice conversion task. We use both full-scale fine-tuning and LoRA fine-tuning. For LoRA fine-tuning, we set the rank $r = 16$, resulting in only **9M trainable parameters** excluding those of the MLP-based adapter. Additionally, we observe that for the voice conversion task, our pre-trained model converges on a single A100 GPU after just 10K steps of LoRA fine-tuning and **5K steps** of full-scale fine-tuning, using randomly sampled **0.4K hours** of training data.

**Evaluation and Baseline** We evaluate our models on the VCTK [75] dataset. we randomly select 200 samples from the dataset as source speech, and for each sample, we randomly select another sample from the same speaker as the prompt speech. We also evaluate the models on other test sets, the results are shown in Appendix D.2. We compare our models with several recent voice conversion models, including HireSpeech++ [21], LM-VC [68], UniAudio [26], and Vevo [74]. More details about these baselines can be found in Appendix B.2.

Table 2: Results on voice conversion task.

| Model | Training Data | WER(↓) | SIM(↑) | DNSMOS(↑) | NISQA(↑) |
|---|---|---|---|---|---|
| *VCTK* | | | | | |
| HierSpeech++ [21] | 2.8K | 4.87 | 0.38 | 3.40 | 3.79 |
| LM-VC [68] | 1.4K | 8.35 | 0.29 | 3.46 | 3.93 |
| UniAudio [26] | 60K | 9.00 | 0.25 | 3.47 | 4.28 |
| Vevo [74] | 60K | **3.48** | 0.38 | 3.47 | 4.30 |
| Metis-VC LoRA 16, cfg = 0.0 | 0.4K | 4.49 | 0.50 | 3.48 | 4.46 |
| Metis-VC LoRA 16, cfg = 2.0 | 0.4K | 7.90 | **0.55** | 3.46 | 4.42 |
| Metis-VC fine-tune, cfg = 0.0 | 0.4K | 6.65 | 0.48 | **3.49** | **4.47** |

[1] For LoRA 16, we train on one A100 GPU for 10K steps. For fine-tuning, we train on one A100 GPU for 5K steps.
[2] The best and the second best result is shown in **bold** and by underlined.

**Result** The results are presented in Table 2. The table shows 1) For similarity, our model outperforms all baseline systems, achieving a SIM score of 0.55, significantly higher than the best baseline result of 0.38. 2) For WER, Metis-VC LoRA 16, cfg = 0.0 achieves a competitive result of 4.49, second only to Vevo. 3) For audio quality, as measured by DNSMOS and NISQA, our model shows competitive performance, with slight improvements over most baseline systems. In addition, our model requires only a significantly small amount of data to fine-tune for voice conversion.

### 4.2.3 Target Speaker Extraction

**Implementation Details** For target speaker extraction, we randomly sample 10K hours of data from the pre-training dataset to create the fine-tuning training set without any filtering. During training, samples are dynamically mixed to simulate multi-speaker speech data. Specifically, a random prefix from a sample is extracted as the prompt, and the remaining portion is mixed with another randomly selected sample by directly adding their waveforms. The `w2v-bert-2.0` features of the mixed speech are then used as conditions. We use two fine-tuning methods: full-scale fine-tuning and LoRA fine-tuning. For LoRA fine-tuning, we explore different LoRA ranks $r = 4, 16, 32$, resulting in LoRA modules with trainable parameters of 2M, 9M, and 18M, respectively.

Table 3: Results on target speaker extraction task.

| Model | WER(↓) | SIG(↑) | BAK(↑) | OVRL(↑) | NISQA(↑) | SIM(↑) |
|---|---|---|---|---|---|---|
| *LibriMix test* | | | | | | |
| Ground Truth | 4.27 | 3.62 | 4.03 | 3.32 | 4.11 | 0.76 |
| UniAudio [26] | 20.08 | 3.64 | **4.15** | 3.33 | 4.32 | 0.66 |
| VoiceFilter [22] | 20.10 | 3.27 | 3.77 | 2.91 | 2.97 | 0.68 |
| WeSep [23] | **6.19** | 3.56 | 3.93 | 3.23 | 4.04 | 0.73 |
| TSELM [76] | 9.20 | 3.55 | 4.08 | 3.29 | 4.03 | 0.27 |
| Metis-TSE LoRA 4 | 13.57 | **3.66** | 4.02 | 3.34 | 4.40 | **0.75** |
| Metis-TSE LoRA 16 | 12.52 | **3.66** | 4.02 | **3.35** | **4.41** | **0.75** |
| Metis-TSE LoRA 32 | 9.65 | **3.66** | 4.02 | **3.35** | 4.38 | **0.75** |
| Metis-TSE fine-tune | 6.31 | 3.65 | 4.02 | 3.34 | 4.36 | 0.74 |
| *EmiliaMix test* | | | | | | |
| Ground Truth | 0.00 | 3.57 | 4.01 | 3.27 | 3.83 | 0.86 |
| UniAudio [26] | 32.51 | 3.55 | 4.03 | 3.20 | 4.01 | 0.55 |
| VoiceFilter [22] | 24.10 | 3.21 | 3.64 | 2.78 | 2.15 | 0.71 |
| WeSep [23] | **5.58** | 3.50 | 3.85 | 3.12 | 3.86 | **0.81** |
| TSELM [76] | 52.12 | 3.48 | **4.05** | 3.20 | 3.87 | 0.26 |
| Metis-TSE LoRA 4 | 11.55 | 3.59 | 3.84 | **3.21** | 4.04 | 0.77 |
| Metis-TSE LoRA 16 | 10.48 | **3.60** | 3.85 | **3.21** | **4.06** | 0.77 |
| Metis-TSE LoRA 32 | 8.72 | **3.60** | 3.86 | **3.21** | 4.05 | 0.78 |
| Metis-TSE fine-tune | 6.87 | **3.60** | 3.85 | **3.21** | **4.06** | 0.78 |

[1] The best and the second best result is shown in **bold** and by underlined.
[2] For LibriMix, we use the original text for computing WER. For EmiliaMix, we use the ASR-transcribed text of the ground truth speech for WER calculation.

**Evaluation and Baseline** We evaluate our models on the LibriMix [77] test set. LibriMix is used exclusively for evaluation and is not included in the training set, thus highlighting the generalization

---

[1] https://github.com/myshell-ai/OpenVoice

capabilities of our models. In addition, we construct a test set of 1K samples from the Emilia dataset, which features greater diversity in both speakers and acoustic environments. We denote it by EmiliaMix test. We compare our models with several open-source expert models for target speaker extraction: WeSep [23], VoiceFilter [22], and TSELM [76]. We also compare with UniAudio [26]. See more details about these baselines in Appendix B.3.

**Result**    The results are presented in Table 3. The Table shows: 1) All fine-tuned models exhibit similar performance on audio quality metrics, while the WER decreases as the number of trainable parameters increases. 2) Our models demonstrate significant improvements across two test sets in audio quality metrics, DNSMOS and NISQA, compared to the baselines. Metis-TSE LoRA 16 achieves state-of-the-art NISQA scores of 4.41 on LibriMix test and 4.46 on EmiliaMix test, surpassing the ground truth scores of 4.11 and 3.83, respectively. 3) Metis-TSE fine-tune achieves a competitive WER, although it is slightly lower than WeSep.

### 4.2.4   Speech Enhancement

**Implementation Details**    For speech enhancement, we simulate noisy speech following previous works [24, 48, 78, 79]. We utilize noise datasets such as WHAM! [80] and DEMAND [81], along with room impulse response (RIR) datasets [82], OpenSLR26 and OpenSLR28, in accordance with the 2020 DNS-Challenge[2]. For clean speech data, we randomly sample 10K hours from our pre-trained dataset without any filtering. Inspired by prior works [48, 78], we simulate speech degradation by probabilistically applying various distortions to the audio signals. These include adding noise within an SNR range of -5 to 20 dB with probability $p = 0.9$, introducing reverberation with $p = 0.35$, and limiting the speech signal bandwidth (randomly selected from 2 kHz, 4 kHz, or 8 kHz) with a probability of with $p = 0.25$. The `w2v-bert-2.0` features extracted from the degraded speech serve as input conditions for our models. We fine-tune the models using both full-scale fine-tuning and LoRA fine-tuning. For LoRA fine-tuning, we also experiment with $r = 4, 16, 32$, corresponding to LoRA modules with trainable parameter counts of 2M, 9M, and 18M, respectively.

**Evaluation and Baseline**    We evaluate our models using the 2020 DNS-Challenge [83] test sets, which consist of three categories: 1) synthetic data with reverb, 2) synthetic data without reverb, and 3) real recordings. For comparison, we downsample the output of our models to 16 kHz. We compare our models with several recent speech enhancement models, including TF-GridNet [24], Voice-Fixer [78], SELM [79] and MaskSR [48]. Notably, MaskSR is a model that also uses masked generative modeling but directly generates acoustic tokens for speech enhancement. See more details about these baselines in Appendix B.4.

Table 4: Results on speech enhancement task.

| Model | SIG(↑) | BAK(↑) | OVRL(↑) | NISQA(↑) | SIM(↑) |
|---|---|---|---|---|---|
| *DNS2020 with reverb* | | | | | |
| TF-GridNet [24] | 3.11 | 3.23 | 2.51 | 2.61 | 0.69 |
| VoiceFixer [78] | 3.43 | 4.02 | 3.13 | 3.82 | 0.91 |
| SELM [79] | 3.16 | 3.58 | 2.70 | - | - |
| MaskSR [48] | 3.53 | 4.07 | 3.25 | - | - |
| Metis-SE LoRA 4 | 3.67 | 4.13 | 3.43 | 4.54 | 0.93 |
| Metis-SE LoRA 16 | 3.67 | 4.13 | 3.43 | **4.57** | **0.94** |
| Metis-SE LoRA 32 | **3.68** | **4.14** | 3.43 | 4.48 | **0.94** |
| Metis-SE fine-tune | **3.68** | **4.14** | **3.44** | 4.56 | **0.94** |
| *DNS2020 no reverb* | | | | | |
| TF-GridNet [24] | 3.54 | 4.05 | 3.27 | 4.35 | 0.68 |
| VoiceFixer [78] | 3.50 | 4.11 | 3.25 | 4.27 | 0.96 |
| SELM [79] | 3.51 | 4.10 | 3.26 | - | - |
| MaskSR [48] | 3.60 | 4.15 | 3.37 | - | - |
| Metis-SE LoRA 4 | 3.65 | 4.15 | 3.43 | **4.82** | 0.97 |
| Metis-SE LoRA 16 | 3.65 | 4.16 | 3.43 | 4.81 | 0.97 |
| Metis-SE LoRA 32 | **3.66** | **4.17** | **3.44** | 4.77 | 0.97 |
| Metis-SE fine-tune | 3.64 | **4.17** | 3.43 | 4.76 | **0.97** |
| *DNS2020 Real Recording* | | | | | |
| VoiceFixer [19] | 3.31 | 3.93 | 3.00 | 3.66 | - |
| SELM [79] | 3.59 | 3.44 | 3.12 | - | - |
| MaskSR [48] | 3.43 | 4.03 | 3.14 | - | - |
| Metis-SE LoRA 4 | **3.60** | 4.02 | 3.29 | 3.94 | - |
| Metis-SE LoRA 16 | **3.60** | **4.04** | **3.30** | **3.97** | - |
| Metis-SE LoRA 32 | 3.59 | 3.99 | 3.26 | 3.92 | - |
| Metis-SE fine-tune | 3.59 | 4.01 | 3.27 | 3.95 | - |

[1] The best and the second best result is shown in **bold** and by underlined.

**Result**    The results are presented in Table 4. The table shows that our models achieve state-of-the-art performance across all benchmarks. 1) Metis-SE achieves state-of-the-art results across all three test sets with the highest SIG, BAK, OVRL, and NISQA scores, showing significant improvements over previous baselines. 2) Results among different fine-tuned versions are comparable, even for Metis-SE LoRA 4. 3) Notably, Metis-SE performs exceptionally well on the real recording dataset, further validating its practical applicability.

[2] https://github.com/microsoft/DNS-Challenge

Table 5: Results on lip-to-speech task.

| Model | WER(↓) | DNSMOS(↑) | NISQA(↑) | SIM(↑) |
|---|---|---|---|---|
| *LRS2* | | | | |
| Lip2Speech-Unit [25] | 33.64 | 3.01 | 2.70 | 29.34 |
| Metis-L2S fine-tune | **32.28** | **3.23** | **3.71** | **59.73** |
| *LRS3* | | | | |
| Lip2Speech-Unit [25] | 38.34 | 2.28 | 1.92 | 32.05 |
| Metis-L2S fine-tune | **31.03** | **3.09** | **3.75** | **56.74** |

Table 6: Results of Metis-Omni on target speaker extraction.

| Model | WER(↓) | SIG(↑) | BAK(↑) | OVRL(↑) | NISQA(↑) | SIM(↑) |
|---|---|---|---|---|---|---|
| *LibriMix test* | | | | | | |
| Ground Truth | 4.27 | 3.62 | 4.03 | 3.32 | 4.11 | 0.76 |
| WeSep [23] | 6.19 | 3.56 | 3.93 | 3.23 | 4.04 | 0.73 |
| TSELM [76] | 9.20 | 3.55 | **4.08** | 3.29 | 4.03 | 0.27 |
| Metis-TSE fine-tune | 6.31 | 3.65 | 4.02 | 3.34 | 4.36 | 0.74 |
| Metis-Omni fine-tune | 5.90 | 3.66 | 4.03 | **3.36** | **4.40** | **0.75** |
| Metis-Omni fine-tune, text-guided | **2.70** | **3.67** | 4.03 | **3.36** | 4.39 | **0.75** |

[1] The best and the second best result is shown in **bold** and by underlined.

### 4.2.5 Lip-to-Speech

**Implementation Details**   We use a combined data set comprising the training and sets of LRW[3], LRS2[4], and LRS3[5] as the training data. We use the same methodology in Ma et al. [84] to pre-process the videos and use the visual speech recognition encoder in Ma et al. [84] to extract visual features that served as the conditions and randomly extract a prefix of the target speech as the prompt.

**Evaluation and Baseline**   We compare our models with Lip2Speech-Unit [85], an expert lip-to-speech model. We use the test sets of LRS2 and LRS3 to evaluate our models following previous works [25, 85]. Notably, over 60% of the samples in the two test sets have durations of less than 2 seconds. Utilizing excessively short audio as a prompt may degrade the sound quality of the inference results, while directly using part of the target audio as a prompt may lead to information leakage. We use speech randomly selected from SeedTTS *test-en* as prompts for each test case.

**Result**   The results are presented in Table 5. The table shows that Metis-L2S outperforms the baseline Lip2Speech-Unit across all metrics. Specifically, the audio quality metrics show substantial improvements and the speaker similarity nearly doubles on both datasets (29.34 → 59.73 on LRS2, 32.05 → 56.74 on LRS3). The model also achieves a lower WER than the baseline.

## 4.3   Multi-Task Fine-Tuning

In addition to fine-tuning the pre-trained model separately for different tasks, we explore the potential of jointly fine-tuning it on multiple tasks, resulting in a multi-task model, which we refer to as Metis-Omni. For this study, we select four tasks: zero-shot TTS, voice conversion, target speaker extraction, and speech enhancement. Further details and experimental results are provided in Appendix E and Table 11.

We take target speaker extraction as an example to illustrate the effectiveness of Metis-Omni in enabling novel applications through task combinations. As shown in Table 6, Metis-Omni outperforms baseline methods in terms of WER. Furthermore, by integrating text-to-speech and target speaker extraction tasks, the text-guided version of Metis-Omni achieves a remarkable WER reduction to 2.70, demonstrating a substantial improvement over all other models. Notably, despite the model not being explicitly trained on text-guided target speaker extraction, it generalizes well to this novel setting, highlighting the advantage of our system in leveraging multimodal conditional inputs to enable flexible and efficient task adaptation.

## 5   Conclusion

In this work, we propose *Metis*, a foundation model for unified speech generation that leverages large-scale unlabeled speech data for pre-training and can be effectively adapted to diverse speech generation tasks through fine-tuning. Our experiments demonstrate that Metis outperforms state-of-the-art task-specific and multi-task systems on zero-shot TTS, voice conversion, target speaker enhancement, speech enhancement, and lip-to-speech after fine-tuning, even with fewer than 20M trainable parameters or up to 300 times less training data for certain tasks while supporting multimodal conditional inputs. In addition, we propose Metis-Omni, a version of our pre-trained model fine-tuned on multiple tasks, which demonstrates further improvements.

---

[3]https://www.robots.ox.ac.uk/~vgg/data/lip_reading/lrw1.html
[4]https://www.robots.ox.ac.uk/~vgg/data/lip_reading/lrs2.html
[5]https://mmai.io/datasets/lip_reading/

# 6 Acknowledgment

The authors gratefully acknowledge the funding support from the National Natural Science Foundation of China (Grant No. 62376237), the Shenzhen Science and Technology Program (Grant No. ZDSYS20230626091302006), the Shenzhen Research Institute of Big Data (Internal Project Fund, Grant No. T00120230002), and the 2023 Shenzhen Stability Science Program. We would also like to thank the anonymous reviewers and the Area Chair for their insightful comments and valuable suggestions, which helped improve our paper.

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

# A  Details of Two Types of Speech Discrete Representations

Metis employs two discrete speech representations for the two-stage generation, following the approach of Wang et al. [12].

1) **SSL tokens**: Derived from SSL features of large-scale speech self-supervised learning models [33, 34, 36, 37], SSL tokens encapsulate both semantic and prosodic information, making them well-suited for conditional generation. To minimize information loss, we employ a vector quantization (VQ) model [53, 54] to quantize SSL features into discrete tokens, in contrast to the k-means approach used in previous works [28, 29, 86]. SSL features are extracted from the 17th layer of `w2v-bert-2.0`[6] [34]. The VQ model has a codebook of 8,192 and a codebook dimension of 8. The model is trained using only reconstruction loss and VQ loss.

2) **Acoustic tokens**: Directly obtained from the waveform via vector quantization. The goal is to preserve all the information from the speech to reconstruct a high-quality waveform. We follow the recipe of DAC codec [87] for model architecture, discriminators, and training losses, with modifications to adopt the Vocos [88] decoder for more efficient training and inference. The 24 kHz speech waveform is compressed into discrete tokens using residual vector quantization (RVQ) across 12 layers, each with a codebook size of 1,024 and a codebook dimension of 8.

# B  Baselines

## B.1  Zero-Shot TTS

**VALL-E [8]**   It uses an AR transformer to predict codes from the first layer of EnCodec [89] and a NAR transformer to predict codes from the remaining layers of EnCodec. We use the released checkpoint in Amphion [90][7] which is pre-trained on 45K hours of the MLS [91] English set.

**VoiceCraft [60]**   A token-filling neural codec language model for text editing and text-to-speech. It predicts multi-layer tokens in a delay pattern. We use the official code and checkpoint[8] which is pre-trained on 9K hours of GigaSpeech [92] dataset.

**CosyVoice [32]**   A two-stage large-scale TTS system. The first stage is an autoregressive model and the second stage is a diffusion model. It is trained on 171K hours of multilingual speech data. We use the official code and checkpoint[9].

**XTTS-v2 [61]**   An open-source multilingual TTS model that supports 16 languages. It is also based on an autoregressive model. We use the official code and checkpoint[10].

**MaskGCT [12]**   An open-source large-scale NAR TTS system that eliminates the need for explicit alignment information between text and speech supervision, as well as phone-level duration prediction. It employs masked generative models for two-stage modeling and is trained on 100K hours of multilingual speech data. We use the official code and checkpoint[11].

**NaturalSpeech 3 [10]**   A large-scale NAR TTS system featuring a factorized speech codec for speech decoupling representation and factorized diffusion models for speech generation. It achieves human-level naturalness on the LibriSpeech test set. We report the scores of LibriSpeech *test-clean* obtained from the original paper.

**VoiceBox [11]**   A large-scale NAR multi-task speech generation model based on flow matching [51]. We report the scores of LibriSpeech *test-clean* obtained from the original paper.

---

[6] https://huggingface.co/facebook/w2v-bert-2.0

[7] https://github.com/open-mmlab/Amphion/tree/main/egs/tts/VALLE_V2

[8] https://huggingface.co/pyp1/VoiceCraft/blob/main/830M_TTSEnhanced.pth

[9] https://huggingface.co/model-scope/CosyVoice-300M

[10] https://huggingface.co/coqui/XTTS-v2

[11] https://github.com/open-mmlab/Amphion/blob/main/models/tts/maskgct

## B.2 Voice Conversion

**HireSpeech++ [21]**    A speech generation system designed based on the VITS architecture [93]. It is trained on 2.8K hours sourced from Libri-light [94] and LibriTTS [95]. We use the official code and checkpoint[12].

**LM-VC [68]**    It uses an AR hierarchical transformer to predict SoundStream [47] codes from soft units similar to HuBERT [33] k-means tokens, trained on the Libri-light dataset [94]. It is not open-source, we report the scores obtained from [74].

**UniAudio [26]**    An AR model system can perform multiple audio generation tasks. It uses 500-cluster k-means tokens from HuBERT [33] to predict their proposed acoustic codes for voice conversion. We use the official code and checkpoint[13]. It uses 60K hours of Libri-light [94] for training voice conversion.

**Vevo [74]**    It is a versatile zero-shot voice imitation model which can control both timbre and style. It is trained on 60K hours of Libri-heavy [96]. We obtain the generated samples from the authors.

## B.3 Target Speaker Extraction

**UniAudio [26]**    An AR model system can perform multiple audio generation tasks including target speaker extraction. We use the official code and checkpoint.

**VoiceFilter [22]**    A system that isolates a target speaker's voice from multi-speaker audio using a reference signal and neural networks for speaker embedding and spectrogram masking. We use the checkpoint [14] provided from Nguyen and Waibel [97].

**WeSep [23]**    A target speaker extraction model trained on LibriMix [77] and VoxCeleb [98]. We use the official code and checkpoint[15].

**TSELM [76]**    A target speaker extraction model that leverages a NAR transformer to predict discrete speech tokens driven from WavLM [37]. We use the official code and checkpoint[16].

## B.4 Speech Enhancement

**TF-GridNet [24]**    A deep neural network for speech separation integrating full- and sub-band modeling in the time-frequency (T-F) domain. It can also used for speech enhancement. We use the checkpoint[17] provided from Interspeech URGENT 2025 Challenge[18].

**VoiceFixer [78]**    A generative framework to address the speech enhancement task. It consists of an analysis stage and a synthesis stage and employs a ResUNet [99] to model the analysis stage and a neural vocoder to model the synthesis stage. We use the official code and checkpoint[19].

**SELM [79]**    A speech enhancement model that leverages an AR transformer to predict discrete speech tokens driven from WavLM [37]. We report the scores obtained from [48].

**MaskSR [48]**    It uses masked generative models to predict acoustic tokens from DAC [87] codec for speech enhancement. We report the scores obtained from the original paper.

---

[12] https://github.com/sh-lee-prml/HierSpeechpp
[13] https://github.com/yangdongchao/UniAudio
[14] https://huggingface.co/nguyenvulebinh/voice-filter
[15] https://huggingface.co/spaces/wenet-e2e/wesep-tse-2speaker-demo/tree/main
[16] https://huggingface.co/Beilong/TSELM
[17] https://huggingface.co/kohei0209/tfgridnet_urgent25/tree/main
[18] https://urgent-challenge.github.io/urgent2025/
[19] https://github.com/haoheliu/voicefixer_main

### B.5 Lip-to-Speech

**Lip2Speech-Unit [25]**    A lip-to-speech model trained to generate multiple targets, mel-spectrograms and quantized self-supervised speech representations. We use the official code and checkpoint[20].

## C    Evaluation Metrics

**SIM**    We evaluate speaker similarity between the prompt speech and the generated speech by computing the cosine similarity of the WavLM TDNN[21][37] speaker embeddings between the generated sample and the prompt. SIM is reported for tasks involving prompt speech, including zero-shot TTS, voice conversion, target speaker extraction, and lip-to-speech. For speech enhancement, we compute SIM between the generated speech and ground truth using a separate checkpoint[22].

**WER**    Word Error Rate measures the intelligibility of the generated speech.    We use `whisper-large-v3`[23] [100] for all languages except Chinese, where we use `paraformer-zh`[24] [101, 102] as the ASR model to calculate WER. WER is reported for zero-shot TTS, voice conversion, target speaker extraction, and lip-to-speech.

**DNSMOS [103]**    DNSMOS is a neural network-based mean opinion score estimator that correlates strongly with human quality ratings. It comprises three components: 1) speech quality (**SIG**), 2) background noise quality (**BAK**), and 3) overall quality (**OVRL**). We report DNSMOS scores for all tasks while providing the average scores for zero-shot TTS and VC tasks, and all three metrics for the remaining tasks.

**NISQA [104]**    NISQA is a deep learning framework for speech quality prediction. We use the public checkpoint[25]. We report NISQA for voice conversion, speech enhancement, target speaker extraction, and lip-to-speech.

**QMOS**    We use the Quality Mean Opinion Score (QMOS) for subjective listening tests to evaluate speech quality. QMOS is rated on a 5-point scale: 5 (Excellent), 4 (Good), 3 (Fair), 2 (Poor), and 1 (Bad).

**SMOS**    We use the Speaker Mean Opinion Score (SMOS) for subjective listening tests to assess speaker similarity between the generated speech and the prompt. SMOS is rated on a 5-point scale: 5 (Excellent), 4 (Good), 3 (Fair), 2 (Poor), and 1 (Bad). SMOS is reported for zero-shot TTS.

---

[20]https://github.com/choijeongsoo/lip2speech-unitn
[21]https://github.com/microsoft/UniSpeech/tree/main/downstreams/speaker_verification
[22]https://huggingface.co/microsoft/wavlm-base-plus-sv
[23]https://huggingface.co/openai/whisper-large-v3
[24]https://huggingface.co/funasr/paraformer-zh
[25]https://github.com/gabrielmittag/NISQA/blob/master/weights/nisqa.tar

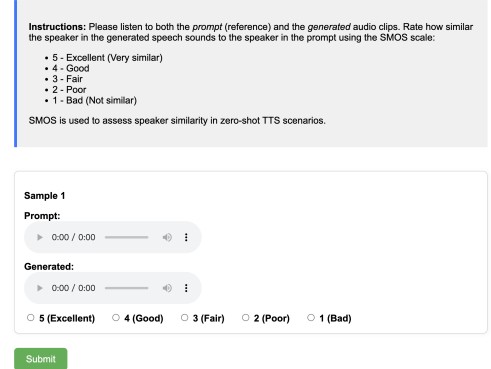

(a) A screenshot of the QMOS evaluation interface.     (b) A screenshot of the CMOS evaluation interface.

Figure 3: Screenshots of the subjective evaluation webpages.

# D Additional Experimental Results

## D.1 Zero-Shot TTS Results on LibriSpeech

Table 7 shows zero-shot TTS results on LibriSpeech *test-clean*. Metis-TTS achieves the highest SIM despite using significantly less training data, while maintaining a competitive WER.

Table 7: Zero-shot TTS results on LibriSpeech.

| Model | Training Data | WER(↓) | SIM(↑) |
|---|---|---|---|
| *LibriSpeech test-clean* | | | |
| VALL-E [10] | 45K EN | 5.90 | 0.50 |
| VoiceCraft [60] | 9K EN | 4.68 | 0.45 |
| NaturalSpeech 3 [10] | 60K EN | **1.94** | 0.67 |
| VoiceBox [10] | 60K EN | 2.03 | 0.64 |
| Metis-TTS | 10K Multi. | 4.33 | **0.70** |

## D.2 Voice Conversion Results on SeedTTS-VC

We also provide voice conversion results on SeedTTS-VC test sets in Table 8. The test sets are provided by [31][26]. Despite being fine-tuned on significantly less training data (0.4K hours compared to 2.8K hours used by HierSpeech++), Metis-VC achieves higher SIM scores on both SeedTTS-VC-en (0.76 vs. 0.56) and SeedTTS-VC-zh (0.63 vs. 0.39), demonstrating its strong capability in preserving speaker identity. Additionally, Metis-VC achieves a higher NISQA score than the baseline model, indicating improved speech quality.

Table 8: Voice conversion results on SeedTTS-VC test sets.

| Model | Training Data | WER(↓) | SIM(↑) | DNSMOS(↑) | NISQA(↑) |
|---|---|---|---|---|---|
| *SeedTTS-VC-en* | | | | | |
| HierSpeech++ [21] | 2.8K | 4.48 | 0.56 | 3.57 | 3.47 |
| Metis-VC fine-tune, cfg = 0.0 | 0.4K | 7.05 | **0.76** | 3.51 | **4.14** |
| *SeedTTS-VC-zh* | | | | | |
| HierSpeech++ [21] | 2.8K | 5.45 | 0.39 | 3.50 | 4.14 |
| Metis-VC fine-tune, cfg = 0.0 | 0.4K | 6.82 | **0.63** | 3.39 | **3.82** |

## D.3 Results of Subjective Evaluation

We conduct subjective listening tests for two representative tasks: zero-shot TTS and speech enhancement. For zero-shot TTS, we randomly sampled 20 test samples from the results of SeedTTS *test-en*. Both QMOS and SMOS are reported for zero-shot TTS. The results are shown in Table 9. For speech enhancement, we randomly sampled 20 test samples from the results of DNS2020 with reverb. QMOS is reported for speech enhancement. The results are shown in Table 10.

---

[26]https://github.com/BytedanceSpeech/seed-tts-eval

Table 9: Results of subjective evaluation for zero-shot TTS.

| Model | QMOS(↑) | SMOS(↑) |
|---|---|---|
| Ground Truth | 4.32 ±0.28 | 4.01 ±0.30 |
| VALL-E [8] | 3.32 ±0.22 | 3.58 ±0.13 |
| VoiceCraft [60] | 3.58 ±0.17 | 3.65 ±0.19 |
| CosyVoice [32] | 4.02 ±0.10 | 3.97 ±0.21 |
| Metis-TTS | **4.21** ±0.31 | **4.19** ±0.18 |

Table 10: Results of subjective evaluation for speech enhancement.

| Model | QMOS(↑) |
|---|---|
| Ground Truth | 4.57 ±0.13 |
| TF-GridNet [24] | 3.65 ±0.35 |
| VoiceFixer [78] | 3.77 ±0.29 |
| Metis-SE | **4.29** ±0.21 |

# E   Metis-Omni: Multi-Task Fine-Tuning

In addition to fine-tuning the pre-trained model separately for different tasks, we explore the potential of jointly fine-tuning a pre-trained model on multiple tasks resulting a multi-task model, which we refer to as **Metis-Omni**. For this study, we select four tasks: zero-shot TTS, voice conversion, target speaker extraction, and speech enhancement. We show more details in Appendix E. A subset of 10K hours of data is randomly sampled from the Emilia dataset as the shared training data for these tasks. During training, the task proportions are set as $p = \{0.5, 0.1, 0.2, 0.2\}$ for zero-shot TTS, voice conversion, target speaker extraction, and speech enhancement, respectively. As shown in Table 11, Metis-Omni achieves performance that is either on par with or surpasses task-specific fine-tuned models across all tasks. The only exception is the WER metric in zero-shot TTS, where performance is slightly lower. A possible reason for this is that, under the same number of training steps, multi-task fine-tuning allocates fewer steps to zero-shot TTS, leading to less task-specific optimization.

Table 11: Results of Metis-Omni on four speech generation tasks.

| Task | Dataset | Model | Performance Metrics | Results |
|---|---|---|---|---|
| Zero-shot TTS | *SeedTTS-en* | MaskGCT [12]
Metis-TTS
Metis-Omni | WER(↓) \| SIM(↑) \| DNSMOS(↑) | 2.47 \| 0.72 \| 3.52
**2.41** \| **0.72** \| **3.57**
4.78 \| 0.71 \| **3.57** |
| | *SeedTTS-zh* | MaskGCT [12]
Metis-TTS
Metis-Omni | WER(↓) \| SIM(↑) \| DNSMOS(↑) | 2.18 \| 0.77 \| 3.58
**2.30** \| **0.77** \| 3.55
4.39 \| **0.77** \| **3.60** |
| VC | *VCTK* | LM-VC [68]
Metis-VC
Metis-Omni | WER(↓) \| SIM(↑) \|DNSMOS(↑) | 8.35 \| 0.29 \| 3.46
6.65 \| **0.48** \| 3.49
**3.52** \| 0.34 \| **3.51** |
| SE | *DNS2020 with reverb* | TF-Grident [24]
Metis-SE
Metis-Omni | SIG(↑) \| BAK(↑) \| OVRL(↑) \| NISQA(↑) | 3.11 \| 3.23 \| 2.51 \| 2.61
**3.68** \| **4.14** \| **3.44** \| **4.56**
**3.68** \| 4.13 \| **3.44** \| 4.53 |
| | *DNS2020 no reverb* | TF-Grident [24]
Metis-SE
Metis-Omni | SIG(↑) \| BAK(↑) \| OVRL(↑) \| NISQA(↑) | 3.54 \| 4.05 \| 3.27 \| 4.35
3.64 \| **4.17** \| 3.43 \| 4.77
**3.65** \| **4.17** \| **3.44** \| **4.79** |
| | *Real Recording* | VoiceFixer [78]
Metis-SE
Metis-Omni | SIG(↑) \| BAK(↑) \| OVRL(↑) \| NISQA(↑) | 3.31 \| 3.93 \| 3.00 \| 3.66
3.59 \| **4.01** \| **3.27** \| 3.95
**3.60** \| 4.00 \| **3.27** \| **3.96** |
| TSE | *LibriMix test* | WeSep [23]
Metis-TSE
Metis-Omni | SIG(↑) \| BAK(↑) \| OVRL(↑) \| NISQA(↑) | 3.56 \| 3.39 \| 3.23 \| 4.04
3.65 \| 4.02 \| 3.34 \| 4.36
**3.66** \| **4.03** \| **3.36** \| **4.40** |
| | *EmiliaMix test* | WeSep [23]
Metis-TSE
Metis-Omni | SIG(↑) \| BAK(↑) \| OVRL(↑) \| NISQA(↑) | 3.50 \| 3.85 \| 3.12 \| 3.86
3.60 \| **3.85** \| 3.21 \| 4.06
**3.62** \| 3.82 \| **3.24** \| **4.11** |

## F  Speech Discrete Representation

Speech representation is a crucial aspect of speech generation. Recently, some speech generation systems [8, 10, 12, 28, 29, 30, 31, 74] have transitioned to using discrete speech representations, which can be broadly categorized into two types: 1) **SSL tokens**: typically derived by quantizing speech self-supervised learning features [33, 34, 35, 36, 37]. Unlike acoustic tokens, SSL tokens are designed not for directly reconstructing waveform but for encoding essential semantic and prosody information in speech, making them more suitable for prediction in conditional generation models. 2) **Acoustic tokens**: typically obtained by training a VQGAN [53, 54] model for waveform reconstruction, as used in speech codecs [47, 87, 89]. Acoustic tokens are effective for reconstructing high-quality waveforms. Currently, some speech generation systems [12, 29, 30, 39] utilize both types of representations for speech generation. In this work, we also adopt this two-stage paradigm, employing an MGM to generate SSL tokens from any condition and another MGM to generate acoustic tokens from SSL tokens.

## G  Limitation and Future Work

There are still some limitations that can be studied in the future.

**Unified Audio Representation**   In this work, we utilize two distinct discrete speech representations, SSL tokens and acoustic tokens, for two-stage modeling, with SSL tokens specifically designed for the speech domain. Developing a unified discrete representation for all audio types, such as speech, music, and sound effects, that can seamlessly integrate with conditional generation models is an important direction for future research. Additionally, unifying the characteristics of SSL tokens and acoustic tokens to enable both high-quality waveform reconstruction and effective conditional modeling is equally significant.

**Few-Shot Task Learning**   In this work, we adapt our pre-trained model to different characters through fine-tuning. In the field of NLP, large language models exhibit the remarkable ability to learn new tasks in a zero-shot manner without requiring additional training. This capability merits further investigation in future research.

## H  Impact Statement

Given that Metis is a powerful foundation model capable of generating high-quality speech across multiple tasks, it also presents potential risks of misuse, such as voice spoofing, speaker impersonation, and unauthorized content generation. To mitigate these risks, it is essential to develop robust detection mechanisms for synthetic speech, establish safeguards to prevent malicious use, and implement a responsible reporting system for identifying and addressing misuse cases.

