# OpenReview forum: "Metis: A Foundation Speech Generation Model with Masked Generative Pre-training"
_NeurIPS.cc/2025/Conference — NeurIPS 2025 poster_

### Official Review · Reviewer_o6KH · 2025-07-02

**Clarity:** 3
**Significance:** 4
**Originality:** 2
**Rating:** 4
**Confidence:** 5

**Summary:**

In this paper, the authors propose a framework that first pre-trains a model using a large amount of unlabeled data, and then fine-tunes it for specific downstream tasks. They introduce different conditioning methods for two categories of downstream tasks: (1) non-frame-level conditioning, and (2) frame-level conditioning. Through evaluation, the authors conduct comprehensive experiments on five different tasks and demonstrate the effectiveness of the pre-trained model and the proposed framework. They also argue that, during the fine-tuning stage, the model can be fine-tuned using a relatively small amount of data and a small number of parameters through LoRA.

**Questions:**

- The proposed method appears to adopt a non-autoregressive generation approach (with iterative decoding), rather than the more common autoregressive Transformer-based methods typically used in the NLP domain. Is there any specific insight or reasoning behind this architectural choice? One possible advantage could be faster inference speed. It would be helpful to know how the inference speed of the proposed method compares to that of other existing approaches.
- Why does the model require a prompt during the pre-training stage? Is it because, for non-frame-level conditioning tasks, the model needs to attend to the conditioning information that appears before the generated sequence? If so, does omitting the prompt during pre-training degrade the model’s performance, even after fine-tuning? It would be helpful if the authors clarified whether prompting in pre-training is essential for enabling effective conditioning during downstream tasks.
- As already mentioned in the weaknesses section, is the choice of LoRA rank related to the difficulty of the tasks? For example, when using LoRA with rank 4 for zero-shot TTS or VC tasks, was the performance significantly degraded or was the model unable to train effectively?
- Additionally, what happens when the tasks are trained with a smaller amount of data? Using 1,000 hours of data is still a large amount—especially for low-resource languages or personalized systems. If this model can demonstrate effectiveness with significantly less data, it would be highly valuable for the research community.

**Ethical Concerns:**

["NO or VERY MINOR ethics concerns only"]

**Final Justification:**

I believe this paper will contribute to the community by proposing a general-purpose speech foundation model using masked generative modeling (MGM) and demonstrating its adaptability to various tasks. Moreover, the authors plan to release the code publicly.

**Limitations:**

Yes

**Paper Formatting Concerns:**

No paper formatting concerns

**Quality:**

4

**Strengths And Weaknesses:**

### Strengths
- This paper demonstrates that the proposed methods can be utilized for various downstream tasks through fine-tuning with a relatively small amount of data and achieve better performance than comparison models across various tasks.
- The authors conduct a comprehensive evaluation on five different tasks: zero-shot TTS, voice conversion, target speaker extraction, speech enhancement, and lip-to-speech. Additionally, they demonstrate the potential of the proposed model for multi-task learning through experiments. This evaluation clearly demonstrates the effectiveness of the proposed method.
- I believe this research will benefit the community by showcasing the potential of a general-purpose speech foundation model. The authors also plan to release the model code and checkpoints along with the paper.
### Weaknesses
- From an application standpoint, the authors conduct comprehensive experiments and achieve strong results. However, in terms of methodological novelty, it is difficult to identify a clearly novel aspect of the proposed approach. The main novelty appears to lie in using different fine-tuning strategies depending on the nature of the downstream task—specifically, prompting for non-frame-level conditioning and adding the condition to the input for frame-level conditioning. However, I believe both conditioning methods are not novel and are commonly used in various large-scale models.
- In the evaluation, the explanation or ablation study regarding architectural choices is limited. For example, the authors use different LoRA ranks for different downstream tasks, but there is no insight or explanation provided for this decision. It remains unclear whether the choice of LoRA rank depends on the difficulty or complexity of the downstream task.
- The purpose of training a general-purpose foundation model is to enable effective and efficient adaptation to a variety of tasks. The authors claim that their proposed method can be fine-tuned for various tasks using a small amount of data. However, in the zero-shot TTS task, they use 1,000 hours of data, which is not a small amount. Therefore, to better demonstrate the effectiveness and generalization ability of their foundation model, the authors should fine-tune it with a smaller dataset and showcase its performance under low-resource conditions.

---

> ### Author Rebuttal · Authors · 2025-07-31
>
> Dear reviewer, we want to thank for your carefully reading and providing the constructive comments! Below we address the concerns mentioned in the review.
>
> `Weakness 1: From an application standpoint, the authors conduct comprehensive experiments and achieve strong results... However, I believe both conditioning methods are not novel and are commonly used in various large-scale models.`
>
> Thank you for the thoughtful feedback! While some components of our method (e.g., conditioning strategies) have been explored in prior work, the core novelty of our approach lies in leveraging masked generative modeling for pretraining a unified foundation model for speech generation. This model can be efficiently adapted to a wide range of speech generation tasks using only a small amount of training data and trainable parameters. The fine-tuning strategies across tasks are distinguished only by the type of condition, whether it is frame-level or not. As a result, tasks such as VC, SE, TSE, and lip2speech share the same fine-tuning and model architecture, differing only in the form of input condition, which naturally reflects their task-specific requirements. We also propose a unified fine-tuning and inference mechanism, **as discussed in our response to Reviewer zbxf (weakness 1), to further simplify and standardize the framework across tasks**.
>
> Our goal is not to solve a single task with masked generative models, but to build a general-purpose foundation model that can flexibly adapt to diverse speech generation scenarios through a shared architecture and training pipeline.
>
> `Weakness 2: In the evaluation, the explanation or ablation study regarding architectural choices is limited...`
>
> We are happy to provide more clarification regarding the choice of LoRA ranks for different tasks. First, we find that LoRA fine-tuning is effective across all tasks, though some tasks can achieve strong performance with significantly fewer trainable parameters.
> In general, higher LoRA ranks lead to better performance. However, for certain tasks, even a very small LoRA rank (e.g., 4) achieves satisfactory results. For example, in SE and TSE, we observe that a LoRA rank of 4 is sufficient. We hypothesize this is because these tasks use frame-level conditions that are closely aligned with the target speech tokens, and the model's primary goal is to extract the target speech from the input.
>
> In contrast, for tasks like TTS, the model must implicitly learn the alignment between the condition (text) and the target speech tokens, which requires more modeling capacity. We find that a LoRA rank of 32 provides strong performance in this setting, while lower ranks lead to notable degradation—particularly in WER. For example, using a LoRA rank of 4 for TTS results in a WER of around 10 on the SeedTTS-en test set. Notably, even with a LoRA rank of 32, the number of trainable parameters remains relatively small, only around 32M.
>
> We will add these details to the paper to provide a clearer explanation of our rank selection strategy.
>
> `Weakness 3: The purpose of training a general-purpose foundation model is to enable effective and efficient adaptation to a variety of tasks... the authors should fine-tune it with a smaller dataset and showcase its performance under low-resource conditions.`
>
> Thank you for the great suggestion! In our experiments, we have explored training with even less than 1000 hours of data. For example, in Section 4.2.1, we fine-tune Metis-TTS LoRA-32 on the LibriSpeech dataset using only 0.58K hours of training data. Despite the small scale, the model achieves a WER of 4.55 on the SeedTTS test-en set, outperforming strong baselines such as VALL-E (trained on 45K hours) and VoiceCraft (9K hours), and approaching the performance of CosyVoice (170K hours). It also achieves a SIM score of 0.72, surpassing most baselines.
>
> Similarly, in the voice conversion (VC) task, we use only 0.4K hours of training data and still outperform the SOTA baseline Vevo (trained on 60K hours) in terms of SIM and quality metrics, with a slightly higher WER (4.49 vs. 3.38).
>
> To further validate the low-data regime setting, we extend our experiments to even smaller training sets. Due to resource and data constraints, we take TTS and VC as representative tasks and randomly sample 0.1K hours of English data from the Emilia dataset for fine-tuning.
>
> SeedTTS en:
> | Model                     | WER  | SIM  |
> |-|-|-|
> | Metis-TTS  finetune 10K | 2.28 | 0.72 |
> | Metis-TTS  finetune 1K    | 3.86 | 0.71 |
> | Metis-TTS   finetune 0.1K  | 5.97 | 0.68 |
>
> VCTK:
> | Model                     | WER  | SIM  |
> |-|-|-|
> | Metis-VC finetune  0.4K    | 6.65 | 0.48 |
> | Metis-VC finetune 0.1K  | 7.01 | 0.45 |
>
> `Question 1: The proposed method appears to adopt a non-autoregressive generation approach... It would be helpful to know how the inference speed of the proposed method compares to that of other existing approaches.`
>
> We are happy to provide more justification for choosing masked generative modeling (MGM) as our modeling approach for speech tokens.
>
> - Previous works have demonstrated that MGM can yield better performance under comparable conditions for speech generation tasks. For instance, [1] shows improvements in TTS, and [2,3] report superior results for speech enhancement compared to AR baselines.
>
> - For most tasks with non-frame-level conditions, such as VC, SE, TSE, the conditioning inputs are typically more dense and locally aligned with the target speech tokens. We observe that the attention matrices in these tasks exhibit strong locality, making MGM a more efficient modeling choice. Additionally, as a NAR method, MGM supports faster inference compared to AR models, as detailed below.
>
> - Compared with AR models, MGM shows advantages in both training and inference efficiency, especially in tasks with frame-level conditions. AR models often require concatenating condition tokens with target tokens or using cross-attention, both of which significantly increase training complexity. In contrast, MGM can model targets independently and requires only a few iterative steps during inference. While AR models can benefit from KV caching, we find that MGM offers better inference speed in most scenarios under the same sequence length. For example, in our TTS experiments, MGM achieves lower RTF (0.35 vs. 0.57), and its advantage becomes even more pronounced as the token rate decreases. Our recent exploration shows that at a token rate of 12.5Hz, MGM achieves a lower RTF (0.18 vs. 0.31) than AR for TTS (The RTF is measured on an H100 GPU using 10-second speech samples).
>
> - MGMs represent a promising and under-explored generative paradigm in speech pretraining. While prior works such as MaskGCT (for TTS) and AnyEnhance (for SE) have adopted MGM, they are task-specific models. In contrast, our goal is to explore the potential of MGM in building a unified pretraining framework for various speech generation tasks. Moreover, recent studies have begun exploring MGM-based pretraining in other domains, such as text generation [4] and text-image multimodal modeling [5], further validating its generality and potential.
>
> `Question 2: Why does the model require a prompt during the pre-training stage?...`
>
> In this work, introducing the prefix prompt mechanism during pretraining is a design choice to enable the model to learn how to utilize prompts for tasks that require them. This mechanism improves the model's adaptability to downstream tasks involving prompts, such as zero-shot TTS, voice conversion, and target speaker extraction. For tasks that do not require prompts, such as speech enhancement, no prompt is used during fine-tuning. In practice, we adopt a mixed strategy during pretraining: prompts are introduced in 80% of training steps, while the remaining 20% are trained without prompts. This design encourages the model to develop in-context learning abilities, specifically to better support prompt-based downstream tasks.
>
> We are happy to provide an ablation study. We retrained the pre-trained model without using prompts, and then fine-tuned it on the zero-shot TTS task with the prompt mechanism. (Due to time and resource constraints, we will include results for other tasks in future work.) The results show that removing prompts during pretraining leads to a drop in SIM, while WER remains largely unaffected.
>
> SeedTTS en:
> | Model                     | WER  | SIM  |
> |-|-|-|
> | Metis-TTS   | 2.28 | 0.72 |
> | Metis-TTS (pre-training no prompt)     | 2.41 | 0.70 |
> | Metis-TTS (without pre-training)     | 4.91 | 0.69 |
>
> SeedTTS zh:
> | Model                     | WER  | SIM  |
> |-|-|-|
> | Metis-TTS   | 2.30 | 0.77 |
> | Metis-TTS (pre-training no prompt)     | 2.59 | 0.73 |
> | Metis-TTS (without pre-training)     | 4.98 | 0.73 |
>
> `Question 3: ...`
>
> We discuss the question in weakness 2.
>
> `Question 4: Additionally, what happens when the tasks are trained with a smaller amount of data?...`
>
> Thank you for the insightful question! We discuss the question in weakness 3.
>
> Thanks again for your valuable comments, we would be grateful if we could hear your feedback regarding our answers. We would be happy to answer and discuss if you have further comments.
>
> Reference:
>
> > [1] Wang Y, Zhan H, Liu L, et al. Maskgct: Zero-shot text-to-speech with masked generative codec transformer[J]. arXiv preprint arXiv:2409.00750, 2024.
>
> > [2] Zhang J, Yang J, Fang Z, et al. Anyenhance: A unified generative model with prompt-guidance and self-critic for voice enhancement[J]. arXiv preprint arXiv:2501.15417, 2025.
>
> > [3] Li X, Wang Q, Liu X. Masksr: Masked language model for full-band speech restoration[J]. arXiv preprint arXiv:2406.02092, 2024.
>
> > [4] Nie S, Zhu F, You Z, et al. Large language diffusion models[J]. arXiv preprint arXiv:2502.09992, 2025.
>
> > [5] Yang L, Tian Y, Li B, et al. Mmada: Multimodal large diffusion language models[J]. arXiv preprint arXiv:2505.15809, 2025.

---

> ### Comment · Reviewer_o6KH · 2025-08-05
>
> I appreciate the authors for their sincere explanation and the additional experiments provided in response to the review. The new results comparing pretraining with and without prompting are particularly interesting. It would be helpful if the authors could include some insights on this point in the manuscript.
>
> Regarding one of my main concerns about the amount of data used for fine-tuning, I still personally believe that 0.1 K hours is quite substantial (for example, the YourTTS was built using around 50 hours of data, even though its performance is much lower than that of current zero-shot TTS systems). The authors argue that other baseline systems require even more data; however, since the proposed model has already been pretrained on a significantly larger dataset, I believe this comparison may not be entirely fair. According to the additional experimental results, the performance degrades considerably when the fine-tuning data is reduced to 0.1K hours, which suggests that the amount of data used in the fine-tuning stage is indeed critical.
>
> As mentioned in the limitations section, NLP foundation models are capable of few-shot learning without any fine-tuning, which indicates that such models capture general domain knowledge. Similarly, I believe that strong performance under low-resource fine-tuning conditions would demonstrate the general knowledge captured by the proposed model.
>
> Given the nature of the task (zero-shot and voice conversion), it might be beneficial to reduce the total amount of fine-tuning data while maintaining the number of speakers. For example, assuming a total of 10K samples, using 10 sentences for each of 1K speakers, rather than 100 sentences for each of 100 speakers, may better leverage the general knowledge acquired during pretraining for the zero-shot TTS task.
>
> I acknowledge that developing a general-purpose speech foundation model is highly challenging, especially given the discrepancies across different tasks, and I believe this work proposes a valuable direction with the use of masked generative modeling (MGM).

---

> > ### Author Response · Authors · 2025-08-07
> >
> > Thank you very much for your thoughtful and constructive feedback.
> >
> > We sincerely appreciate your recognition of our work and the valuable suggestions regarding the fine-tuning data regime. We fully agree that demonstrating strong performance under low-resource conditions would better reflect the generalization capability of a pretrained foundation model. As suggested, we will explore more extreme low-resource settings, such as maintaining speaker diversity while reducing per-speaker data to provide deeper insights into the model's capacity for zero-shot TTS and voice conversion.
> >
> > Once again, thank you for your encouraging comments and helpful suggestions.

---

### Official Review · Reviewer_c6Ub · 2025-07-02

**Clarity:** 3
**Significance:** 3
**Originality:** 2
**Rating:** 5
**Confidence:** 3

**Summary:**

This paper proposes a unified speech foundation model based on masked speech modeling and the model is tunable for various speech tasks such as voice conversion, target speaker extraction, TTS, speech enhancement and lip-to-speech. The system uses both self-supervised speech embeddings and audio embeddings. These two sets of input features serve two separate purposes, 1) to map input prompt and 2) to generate high quality audio. Once the masked pre-training is done, there is a fine-tuning stage for the tasks mentioned above. During fine-tuning, the SSL tokens are appended with the conditions (text, audio prompt, lip features). Experiments show results for 5 tasks for both LoRA adapted and finetuned models. The models are compared to the ground truth audio as well as some high-performing models in the literature depending on the task. Outputs are evaluated using scores like WER, speaker similarity, DNSMOS as applicable. A second set of experiments evaluate the case where the finetuning stage involves multiple tasks at once. In that experiment, the authors show that the multitask version of the model can generalize to other tasks such as the text-guided target speaker extraction task. This multitask version slightly outperforms the model tuned only for the TSE task but in terms of WER, it cannot outperform the model trained for text-guided TSE.

**Questions:**

1) Please comment on whether reference [12] has been already published in a peer-reviewed venue.

2) In some experiments, DNSMOS and NISQA evaluations have been used. However, these scores are discriminative enough to compare the quality of various systems. The values themselves may suggest that the system is working but they do not necessarily imply that the proposed model is significantly outperforming another one (for example, Vevo vs. Metis in Table 2). Could you please comment on this? Have you considered using other speech evaluation metrics such as the UTMOSv2 scores?

3) Minor mistake: Section 4.2.2. mentions "HireSpeech++" in the text which should be HierSpeech++ as mentioned in Table 2.

**Ethical Concerns:**

["NO or VERY MINOR ethics concerns only"]

**Final Justification:**

I would like to thank the authors for responding to my concerns especially regarding the novelty of the paper. If the code is also going to be released, I am willing to increase my score to accept.

**Limitations:**

Limitations have been mentioned in the appendix rather than in the main text.

**Quality:**

2

**Strengths And Weaknesses:**

Strengths:

- Quality: A generic speech model training framework has been proposed that can be tuned for various speech tasks. There are various experiments showing positive results as compared to previously published papers.

- Significance: Speech generative modeling is relevant to the speech community and the ability of the proposed model to handle different speech tasks is an important feature to have as a unified model.

Weaknesses:

- Quality: The paper (the text) might be lacking details in terms of reproducibility (even though the author checklist suggests that the code will be made available). Some of the evaluation criteria may not be well-suited for the individual task and/or they do not show demonstrate the model differences well.

- Originality: Based on the text, it seems that the current paper follows reference [12] quite closely with some differences on the text embedding component. If the reference [12] has been published in a peer-reviewed venue, it may make the novelty of this paper somewhat limited. Based on the current pdf, reference [12] is an arxiv paper. If that is still the case, this paper might satisfy the novelty criterion.

- Clarity: It is good to have multiple experiments on various tasks but it might have been better to share more details on the core model and its training details.

---

> ### Author Rebuttal · Authors · 2025-07-31
>
> Dear reviewer, we want to thank for your carefully reading and providing the constructive comments! Below we address the concerns mentioned in the review.
>
> `Weakness 1: Quality: The paper (the text) might be lacking details in terms of reproducibility (even though the author checklist suggests that the code will be made available). Some of the evaluation criteria may not be well-suited for the individual task and/or they do not show demonstrate the model differences well.`
>
> Thank you for the comments! We have made every effort to improve the reproducibility of our work. We plan to open-source the code for each model, along with detailed configuration files specifying hyperparameter settings and model architecture.
>
> For evaluation, all baseline models are open-source and evaluated on public benchmarks. We will also provide scripts for computing evaluation metrics on each test set.
>
> `Weakness 2: Originality: Based on the text, it seems that the current paper follows reference [12] quite closely with some differences on the text embedding component. If the reference [12] has been published in a peer-reviewed venue, it may make the novelty of this paper somewhat limited. Based on the current pdf, reference [12] is an arxiv paper. If that is still the case, this paper might satisfy the novelty criterion.`
>
> Reference [12], MaskGCT: Zero-Shot Text-to-Speech with Masked Generative Codec Transformer, has already been published at ICLR 2025. We cite MaskGCT in multiple parts of our paper, as both our work and MaskGCT adopt masked generative modeling as the core generative paradigm.
>
> However, our contributions go beyond MaskGCT in several key aspects:
>
> - First, while MaskGCT focuses solely on zero-shot TTS, our work extends masked generative modeling to a broader set of speech generation tasks, including zero-shot TTS, voice conversion (VC), speech enhancement (SE), target speaker extraction (TSE), and a multimodal task, lip-to-speech generation.
>
> - Our goal is not to apply masked generative models to solve a single task, but rather to build a unified foundation model with large-scale pretraining, capable of efficiently adapting to a wide range of speech generation tasks. To this end, we design a unified pretraining mechanism, which enables efficient fine-tuning across diverse speech generation tasks. In the pretraining stage, we adopt unconditional masked generative modeling as the objective, without relying on any task-specific conditioning.
>
> Masked generative modeling has shown effectiveness across various domains, e.g., MaskGIT (image generation), MaskGVIT (video generation) in vision, and MaskGCT (TTS), NaturalSpeech 3 (TTS), AnyEnhance (SE and TSE), MaskSR (SE), and in audio, but all of these are task-specific models. In contrast, our work aims to build a unified framework that aims to generalize masked generative modeling for diverse speech generation tasks.
>
> `Weakness 3: Clarity: It is good to have multiple experiments on various tasks but it might have been better to share more details on the core model and its training details.`
>
> We provide the details of the pre-training process in Section 4.1, including the model architecture, dataset, and training strategy. For each downstream task, we describe any differences from the pre-training setup in the corresponding experimental sections. Due to space limitations, we are unable to provide more details in the main text. However, we will do our best to include additional information, including detailed model architecture, in the appendix.
>
> `Question 1: Please comment on whether reference [12] has been already published in a peer-reviewed venue.`
>
> We discuss this question in Weakness 2.
>
> `Question 2: In some experiments, DNSMOS and NISQA evaluations have been used. However, these scores are discriminative enough to compare the quality of various systems. The values themselves may suggest that the system is working but they do not necessarily imply that the proposed model is significantly outperforming another one (for example, Vevo vs. Metis in Table 2). Could you please comment on this? Have you considered using other speech evaluation metrics such as the UTMOSv2 scores?`
>
> Yes, since DNSMOS and NISQA are commonly used for evaluating SE and TSE tasks, in our work, we also adopt DNSMOS for evaluating zero-shot TTS and VC, and additionally use NISQA for VC, in order to maintain a relatively streamlined evaluation setup.
> However, we note that these metrics are not always reliable. For example, DNSMOS and UTMOS are often inaccurate on non-English speech. We present these metrics mainly for reference, and for zero-shot TTS and VC we believe SIM and WER are more meaningful.
> In addition, based on our experience, SIM computed with WavLM embeddings also correlates with speech quality to some extent. We also provide subjective evaluation results in Appendix D.3 Results of Subjective Evaluation.
>
> We are happy to supplement the UTMOS scores below for reference:
>
> Zero-shot TTS:
>
> | Model       | UTMOS (SeedTTS test-en) | UTMOS (SeedTTS test-zh) |
> |-------------|--------------------------|--------------------------|
> | Metis-TTS   | 3.82                     | 2.75                     |
> | MaskGCT     | 3.59                     | 2.68                     |
> | CosyVoice   | 3.78                     | 2.81                     |
> | F5-TTS      | 3.51                     | 2.67                     |
>
> VC:
>
> | Model       | UTMOS (SeedTTS-VC-en) | UTMOS (SeedTTS-VC-zh) |
> |-------------|------------------------|------------------------|
> | Metis-VC    | 3.65                   | 2.97                   |
> | SeedVC      | 3.31                   | 2.69                   |
> | Vevo        | 3.51                   | 2.86                   |
>
> `Question 3: Minor mistake: Section 4.2.2. mentions "HireSpeech++" in the text which should be HierSpeech++ as mentioned in Table 2.`
>
> Thank you for pointing this out!  We will address and fix it.
>
> Thanks again for your valuable comments, we would be grateful if we could hear your feedback regarding our answers. We would be happy to answer and discuss if you have further comments.

---

> > ### Comment · Reviewer_c6Ub · 2025-07-31
> >
> > I would like to thank the authors for responding to my concerns especially regarding the novelty of the paper. If the code is also going to be released, I am willing to increase my score to weak accept.

---

> > > ### Author Response · Authors · 2025-08-01
> > >
> > > Thank you for your thoughtful follow-up and your willingness to reconsider your score. We appreciate your recognition of our clarifications regarding the novelty of the work. We confirm that we will release the code and model checkpoints as soon as the anonymity constraints are lifted, regardless of the final decision, to facilitate future research.

---

### Official Review · Reviewer_vbjN · 2025-07-03

**Clarity:** 3
**Significance:** 3
**Originality:** 2
**Rating:** 5
**Confidence:** 4

**Summary:**

This paper proposes a novel pre-training and fine-tuning framework that integrates self-supervised learning (SSL) with acoustic token representations. The core approach is based on masked pre-training and is extended to support a variety of speech generation tasks by conditioning on task-specific prompts implemented as prefix tokens. Fine-tuning is also conducted using a masked generation strategy. The experiments cover a broad range of tasks, including text-to-speech (TTS), voice conversion (VC), speech enhancement (SE), and target speaker extraction (TSE), demonstrating state-of-the-art performance with a relatively small number of trainable parameters.

**Questions:**

- Abstract: The abstract should be revised for clarity and informativeness. It states that the model uses “fewer than 20M trainable parameters or 300 times less training data,” but fails to specify which baseline models are being compared. Without that context, the claim is vague and less impactful.
- Lines 30–33: The discussion of limitations in prior work is unclear. The claims about the need for large training data or difficulty in task extension lack supporting evidence or citations. Please elaborate with concrete examples, experimental results, or references that justify these limitations, and explain why the proposed method addresses them more effectively.
- Section 3.4: Please clarify the properties of the prompt used in the method. Is the prompt length fixed across tasks? If not, how does it vary (e.g., longer or shorter depending on the task)?
- Section 3.5: The explanation of prompts and conditioning is difficult to follow. Please provide more concrete examples (e.g., what do the prompt and condition look like for TTS, VC, or TSE?) to make the mechanism and design choices clearer.
- Section 4.1: The source of the “200K hours” of training data is not clearly explained. Please provide more details on the datasets used.
- Section 4.2: The experimental setup is hard to interpret because each baseline uses different training datasets, strategies, and model sizes. A more standardized comparison or ablation study would help clarify where the proposed method truly excels.
- Section 4.2.1: I could not find a discussion of how the length parameter $n$ is selected for TTS experiments. Please explain how to set the length.
- Section 4 – Evaluation Metrics: Please reconsider the use of DNSMOS for tasks like TTS and VC, as it is tailored for speech enhancement. Metrics such as UTMOS or other TTS-specific measures would be more appropriate. Subjective evaluations would also help validate perceptual quality. For SE and TSE tasks, reference-based metrics like PESQ and STOI should be included to assess signal-level fidelity.

**Ethical Concerns:**

["NO or VERY MINOR ethics concerns only"]

**Final Justification:**

The additional explanations and ablation study discussions address my concerns, and I now recommend this paper for acceptance.

**Limitations:**

While the paper addresses technical limitations in unified audio representation and few-shot learning capabilities, it lacks a discussion on the broader societal impacts of the work.

**Paper Formatting Concerns:**

No major issues.
However, some of the experimental discussions rely on results presented in the appendix. While this is acceptable, it would be more helpful if the main text provided a more self-contained discussion, allowing readers to follow the key findings without needing to consult supplementary material.

**Quality:**

3

**Strengths And Weaknesses:**

Strengths
- Simple pre-training techniques based on a prompt, which can be applied to various speech generation tasks
- Shows SOTA in various speech generation tasks

Weaknesses
- The idea is too simple and incremental. It is only based on the prompt. Pre-text-based approach is very good, but many speech LLM-based systems are conditioned on a prefix.
- The paper does not have ablation experiments, and its validation is not obvious. E.g., pre-training w/ and w/o prefix would be very informative to show the effectiveness of the proposed method.

---

> ### Author Rebuttal · Authors · 2025-07-31
>
> Dear reviewer, we want to thank for your carefully reading and providing the constructive comments! Below we address the concerns mentioned in the review.
>
> `Weakness 1: The idea is too simple and incremental... but many speech LLM-based systems are conditioned on a prefix.`
>
> In this work, **introducing the prefix prompt mechanism during pretraining is not essential, but rather a design choice to enable the model to learn how to utilize prompts for tasks that require them**. This mechanism improves the model's adaptability to downstream tasks involving prompts, such as zero-shot TTS, voice conversion, and target speaker extraction. For tasks that do not require prompts, such as speech enhancement, no prompt is used during fine-tuning.
>
> We also considered **removing the prefix during pretraining**. 20% of the training steps do not include any prompt, as mentioned on line 216. We emphasize that prefix prompting is not the core focus of our work; it can be introduced solely at the fine-tuning stage for specific tasks if desired. Nonetheless, including it during pretraining does not harm performance, and we adopt the mixed strategy.
>
> `Weakness 2: The paper does not have ablation experiments, and its validation is not obvious. E.g., pre-training w/ and w/o prefix would be very informative to show the effectiveness of the proposed method.`
>
> We conducted ablation studies to isolate the core contribution of our method and validate the effectiveness of the large-scale pretraining–then–fine-tuning paradigm. Taking zero-shot TTS as an example, Metis-TTS with pretraining significantly outperforms Metis-TTS without pretraining on both SIM and WER metrics using the same training data (WER: 2.28 vs. 4.91; SIM: 0.72 vs. 0.69). We also show that with pretraining, our models can be fine-tuned using significantly fewer trainable parameters via LoRA.
>
> We add the ablation study on **with or without prompt for pretraining** in the response to **reviewer o6KH's question 2**.
>
> `Question 1: Abstract: The abstract should be revised for clarity and informativeness... the claim is vague and less impactful.`
>
> Thank you for your suggestion. The "300 times less" refers to the fact that, compared to the 300K hours of pretraining data, our fine-tuning only requires 1K hours or less. We will revise and clarify this in the paper to ensure clearer expression.
>
> `Question 2: Lines 30–33: The discussion of limitations in prior work is unclear...`
>
> Thank you for your suggestion. We have provided clearer evidence to support our claims. For task-specific models, state-of-the-art systems in zero-shot TTS typically rely on large-scale training data, ranging from 60K to over 100K hours (e.g., MaskGCT, CosyVoice, F5-TTS, VoiceBox, and NaturalSpeech 3). Similarly, recently proposed state-of-the-art VC models such as Vevo (ICLR 2025) are trained on 60K hours of data. Multi-task models like UniAudio use a total of 165K hours, with 60K/60K/20K/10K hours allocated to TTS, VC, SE, and TSE respectively, and contain up to 1B trainable parameters.
>
> In contrast, our experiments demonstrate that comparable or even better performance can be achieved with significantly less task-specific fine-tuning data with our masked generative pre-training. For example, in the TTS task, our model achieves performance on par with or better than MaskGCT and CosyVoice (both trained on about 100K hours), using only 10K or even 1K hours of data. In the VC task, we obtain better results than Vevo (SIM 0.50 vs. 0.38), using just 0.4K hours of training data compared to their 60K hours.
>
> Our method addresses this challenge more effectively. It is motivated by the observation that models trained on large-scale data can recover masked SSL tokens from prompts and unmasked tokens, even without task-specific conditions, as shown in other domains [1, 2]. Empirically, our pretrained model is able to generate speech that captures the prosodic style and timbre of the prompt. However, without semantic guidance, the generated speech may lack intelligibility, often resulting in random word concatenations. This highlights the importance of task-specific conditioning, which can be efficiently introduced through fine-tuning, allowing the model to adapt flexibly to various speech generation tasks.
>
> `Question 3: Section 3.4: Please clarify the properties of the prompt used in the method...`
>
> During pretraining, we do not incorporate any task-specific assumptions. For each training sample, we randomly select a prefix of the sequence to serve as a prompt, which remains unmasked with a probability of $p = 0.8$. The prompt length is uniformly sampled from the range [0%, 40%] of the total sequence length.
>
> During finetuning, prompts are used only for tasks that require them (e.g., zero-shot TTS, voice conversion). For tasks that do not require prompts, such as speech enhancement, no prompt is used. In all cases, the prompt mechanism used during fine-tuning is kept consistent with that of pretraining.
>
> `Question 4: Section 3.5: The explanation of prompts and conditioning is difficult to follow...`
>
> We are happy to provide more details for clearer explanation. For all tasks that require prompts, the prompts refer to the SSL tokens of the prompt speech. Specifically, for zero-shot TTS, VC, and TSE, the prompt corresponds to the SSL tokens of the reference speaker's speech. The condition varies depending on the task: for **TTS**, it refers to the text embedding derived from the input text; for **VC, TSE, and SE**, it corresponds to the speech features of the input speech, such as the source speech for VC, the mixed speech for TSE, and the noisy speech for SE. These features can be either mel-spectrograms or SSL features. In this work, we use w2v-BERT 2.0 SSL features, which are temporally aligned with the target SSL tokens, that is, they share the same sequence length along the time axis.
>
> `Question 5: Section 4.1: The source of the “200K hours” of training data is not clearly explained...`
>
> The additional 200K hours of data come from Emilia-Large [3], which primarily consists of YODAS2 [4] data processed through the Emilia pipeline.
>
> `Question 6: Section 4.2: The experimental setup is hard to interpret...`
>
> Comparing with open-source models can be challenging due to differences in experimental settings. However, for some tasks such as TTS and VC, certain baselines are trained using the Emilia dataset, and our fine-tuning data is a randomly selected subset of the same dataset, albeit with significantly less data. For the lip-to-speech task, we use the same training data as the baseline models.
>
> For internal comparisons within our method, such as varying the fine-tuning data size or the number of trainable parameters, we ensure that all other variables are held constant. We appreciate your suggestion and will conduct more rigorous and controlled experiments in future work.
>
> `Question 7: Section 4.2.1: I could not find a discussion of how the length parameter n...`
>
> I would like to clarify your question: does the length parameter n refer to the target speech length during testing? That is also our understanding. Since our method does not rely on explicit alignment between text and speech or a duration predictor, we need to specify a target length at inference time. Following previous NAR TTS models that similarly do not use duration predictors (e.g., MaskGCT, E2TTS, F5-TTS), we adopt a simple rule-based approach to determine the target length:
>
> ```
> target len. = (prompt len. / prompt phone count) × target phone count.
> ```
>
> `Question 8: Section 4 – Evaluation Metrics..`
>
> - Thank you for your suggestion! Due to space limitations, we provide **subjective evaluation results for zero-shot TTS and speech enhancement in Appendix D.3 Results of Subjective Evaluation**, taking these two tasks as representative cases. We evaluate QMOS and SMOS to assess speech quality and speaker similarity, respectively.
>
> - We also provide the results of **UTMOS for TTS and VC in response to Reviewer c6Ub's Question 2**.
>
> - We do not include signal-level metrics such as PESQ in the main paper, as we find these metrics are not well-suited for evaluating generative-model-based methods. This observation is consistent with prior works [5, 6]. We add these results and will include them in the appendix of the paper.
>
> Model | PESQ | ESTOI
> -|-|-
> AnyEnhance | 2.78 | 0.84
> Metis-SE |2.91 | 0.86
>
> Thanks again for your valuable comments, we would be grateful if we could hear your feedback regarding our answers. We would be happy to answer and discuss if you have further comments.
>
> Reference
>
> > [1] Devlin J, Chang M W, Lee K, et al. Bert: Pre-training of deep bidirectional transformers for language understanding[C]//Proceedings of the 2019 conference of the North American chapter of the association for computational linguistics: human language technologies, volume 1 (long and short papers). 2019: 4171-4186.
>
> > [2] He K, Chen X, Xie S, et al. Masked autoencoders are scalable vision learners[C]//Proceedings of the IEEE/CVF conference on computer vision and pattern recognition. 2022: 16000-16009.
>
> > [3] He H, Shang Z, Wang C, et al. Emilia: A large-scale, extensive, multilingual, and diverse dataset for speech generation[J]. arXiv preprint arXiv:2501.15907, 2025.
>
> > [4] Li X, Takamichi S, Saeki T, et al. Yodas: Youtube-oriented dataset for audio and speech[C]//2023 IEEE Automatic Speech Recognition and Understanding Workshop (ASRU). IEEE, 2023: 1-8.
>
> > [5] Li X, Wang Q, Liu X. Masksr: Masked language model for full-band speech restoration[J]. arXiv preprint arXiv:2406.02092, 2024.
>
> > [6] Zhang J, Yang J, Fang Z, et al. Anyenhance: A unified generative model with prompt-guidance and self-critic for voice enhancement[J]. arXiv preprint arXiv:2501.15417, 2025.

---

> > ### Comment · Reviewer_vbjN · 2025-08-05
> >
> > Thanks for your detailed comments.
> > Additional explanations and experimental discussions mitigate my concerns.
> > I want to increase my score.

---

### Official Review · Reviewer_zbfX · 2025-07-03

**Clarity:** 3
**Significance:** 3
**Originality:** 3
**Rating:** 5
**Confidence:** 3

**Summary:**

The paper proposes Metis, a unified speech generation foundation model trained via masked generative pre-training on unlabeled speech data. Metis is trained using two discrete speech representations and can be fine-tuned with minimal labeled data and parameters for multiple downstream tasks. The Key features include: (1) two-stage speech generation using SSL tokens (semantic/prosodic) and acoustic tokens (waveform-level); (2) pre-training on 300K hours of speech with a masked generative model, without any task conditions; (3) efficient adaptation to tasks like zero-shot TTS, voice conversion, speech enhancement, target speaker extraction, and lip-to-speech, using LoRA or full fine-tuning. The model outperforms state-of-the-art systems on multiple tasks and shows strong generalization with limited data.

**Questions:**

(1) Can the authors further unify inference logic so that new tasks don’t require any architectural changes?

(2) How does Metis handle long-form or streaming generation tasks?

(3) How sensitive is the model to the length or content of prompts used in zero-shot or few-shot generation?

**Ethical Concerns:**

["NO or VERY MINOR ethics concerns only"]

**Limitations:**

Yes.

**Quality:**

4

**Strengths And Weaknesses:**

Strengths:

(1) Unified framework: The paper proposes a clear and modular two-stage pipeline (SSL token prediction → acoustic generation) that enables generalization across tasks.

(2) Efficient adaptation: The model achieves SOTA or near-SOTA performance on five speech generation tasks using <20M parameters and orders-of-magnitude less training data.

(3) Multimodal support: The model is capable of handling text, audio, and video conditions, allowing for novel combinations like text-guided speaker extraction (Metis-Omni).

(4) Extensive experiments: Includes strong baseline comparisons, ablations (e.g., with and without pretraining), LoRA configurations, and results on real-world and synthetic test sets.

Weaknesses:

(1) No unified inference pipeline: While pre-training is unified, fine-tuning still requires task-specific configurations (e.g., frame-level vs. non-frame-level conditioning logic).

(2) Lack of cross-task analysis: While Metis-Omni is introduced for multi-task fine-tuning, detailed cross-task interference analysis is not provided (e.g., does speech enhancement degrade TTS?).

---

> ### Author Rebuttal · Authors · 2025-07-30
>
> Dear reviewer, we want to thank for your carefully reading and providing the constructive comments! Below we address the concerns mentioned in the review.
>
> `Weakness 1:  No unified inference pipeline: While pre-training is unified, fine-tuning still requires task-specific configurations (e.g., frame-level vs. non-frame-level conditioning logic).`
>
> The unified fine-tuning and inference logic for both frame-level and non-frame-level conditions indeed simplifies and streamlines the overall pipeline. In our work, the non-frame-level conditional task mainly refers to the TTS task, while all remaining tasks (voice conversion, speech enhancement, target speaker extraction, and lip2speech) rely on frame-level conditions.
>
> To handle non-frame-level conditions, we prepend the condition embeddings to the speech representation sequence along the time axis and apply self-attention over the combined sequence. Recent studies [1,2,3] have shown that for non-autoregressive TTS systems, it is effective to **directly concatenate text embeddings with speech representations at the feature/channel level without requiring upsampling or alignment**. Since text embeddings are much shorter than speech representations in sequence length, **the remaining positions can simply be filled with padding embeddings**.
>
> This approach allows us to unify both frame-level and non-frame-level conditional tasks: *we concatenate the task-specific condition with the speech representations along the feature/channel dimension, rather than the time dimension, and pad the condition to match the sequence length when necessary*.
>
> `Weakness 2: Lack of cross-task analysis: While Metis-Omni is introduced for multi-task fine-tuning, detailed cross-task interference analysis is not provided (e.g., does speech enhancement degrade TTS?).`
>
> Thank you for the suggestion. We will provide more details about the inference process of Metis-Omni.
>
> We observe that Metis-Omni achieves performance that is on par with or even surpasses task-specific fine-tuned models for speech enhancement and target speaker extraction. One exception is the WER metric in zero-shot TTS, where performance is slightly lower. A possible explanation is that, under the same number of training steps, multi-task fine-tuning allocates fewer effective steps to the zero-shot TTS task, resulting in less task-specific optimization.
>
> Additionally, regarding the inference process: All frame-level conditional tasks (i.e., voice conversion, speech enhancement, and target speaker extraction) share the same adapter, and no additional task ID is required during inference. For zero-shot TTS, the inference process is identical to that of Metis-TTS, and the adapter used for frame-level conditional tasks is not activated during TTS inference.
> Metis-Omni also supports some generalized inference patterns, even if they were not explicitly encountered during training. For example, we can perform text-guided inference for target speaker extraction and speech enhancement by providing the transcript of the target speech (obtained via an ASR system) as an additional condition. As shown in Table 6, this leads to improved performance in some cases.
>
> `Question 1: Can the authors further unify inference logic so that new tasks don’t require any architectural changes?`
>
> We propose a unified inference logic for both frame-level and non-frame-level conditional tasks, **as raised in Weakness 1**, by concatenating the task-specific condition with the speech representations along the feature (channel) dimension rather than the time dimension, and padding the condition as needed to match the sequence length.
>
> `Question 2: How does Metis handle long-form or streaming generation tasks?`
>
> In this work, we primarily focus on non-autoregressive (NAR) speech generation tasks and design our model for non-streaming scenarios. However, recent studies [4] have also demonstrated the potential of masked generative models in streaming inference. Specifically, the speech can be split into chunks, where masked generative modeling is applied within each chunk, and autoregressive generation is performed across chunks. We will explore these directions in future work.
>
> `Question 3: How sensitive is the model to the length or content of prompts used in zero-shot or few-shot generation?`
>
> Since our models are trained on speech segments shorter than 30 seconds (due to limitations of the Emilia dataset), we recommend that the prompt length not exceed 15 to 20 seconds. During training, the maximum prompt length was set to half of the total input length. However, we believe this constraint is largely dependent on the training data, and longer prompts could be supported if trained accordingly.
>
> Thanks again for your valuable comments, we would be grateful if we could hear your feedback regarding our answers. We would be happy to answer and discuss if you have further comments.
>
> Reference
>
> > [1] Eskimez S E, Wang X, Thakker M, et al. E2 tts: Embarrassingly easy fully non-autoregressive zero-shot tts[C]//2024 IEEE Spoken Language Technology Workshop (SLT). IEEE, 2024: 682-689.
>
> > [2] Chen Y, Niu Z, Ma Z, et al. F5-tts: A fairytaler that fakes fluent and faithful speech with flow matching[J]. arXiv preprint arXiv:2410.06885, 2024.
>
> > [3] Ning Z, Chen H, Jiang Y, et al. DiffRhythm: Blazingly fast and embarrassingly simple end-to-end full-length song generation with latent diffusion[J]. arXiv preprint arXiv:2503.01183, 2025.
>
> > [4] Arriola M, Gokaslan A, Chiu J T, et al. Block diffusion: Interpolating between autoregressive and diffusion language models[J]. arXiv preprint arXiv:2503.09573, 2025.

---

### Note · Authors · 2025-08-12

Dear ACs, SACs, and Reviewers,

We sincerely thank all reviewers for their invaluable feedback and constructive suggestions. The discussions during the rebuttal period have greatly helped us improve our manuscript. We also appreciate the efforts of the ACs and SACs in organizing and coordinating the review process.

Below is a brief summary of the key points addressed during the rebuttal discussions:

1. **Clarification of Novelty and Core Contribution:** Several reviewers initially raised questions about the methodological novelty. In our rebuttal, we clarified that our core contribution is **building a unified pre-training and fine-tuning paradigm for a broad range of speech generation tasks**. We distinguished our work from task-specific models like MaskGCT by highlighting Metis's general-purpose, unconditional pre-training and its ability to serve as a foundation model.

2. **Extensive New Experiments on Low-Resource Fine-Tuning:** A key concern raised by some reviewers was whether the model was truly effective with a "small amount" of fine-tuning data. To address this, we conducted and presented new experiments fine-tuning Metis on smaller datasets (0.58K, 0.4K, and 0.1K hours). The results demonstrated that Metis maintains strong performance, validating its efficiency and the effectiveness of the pre-training.

3. **Additional Ablation Studies and Evaluation Metrics:** We provided new ablation studies, including the **impact of pre-training with and without prompts**, and a direct **comparison to a model trained without any pre-training**. These studies empirically validated our design choices. Furthermore, to address **evaluation concerns** from reviewers, we supplemented our results with additional metrics (UTMOS, PESQ, STOI).

4. **Unified Inference Pipeline:** Some reviewers asked if task-specific branching in inference could be avoided. We described a unified inference procedure that supports all tasks via feature-dimension concatenation with padding, eliminating the need for separate logic when adding conditions for frame-level and non-frame-level tasks.

Finally, we confirm that we will open-source all of our code (including inference and training) and model weights once the anonymity period ends, in order to contribute to the community.  We once again thank all the constructive suggestions!

---

### Decision · Program_Chairs · 2025-09-17

**Decision:**

Accept (poster)

**Comment:**

This is a paper about building a strong foundation model for various speech generation tasks. There is very little novelty (as pointed out by reviewer vbjN, c6Ub, and o6KH). The prompts during pre-training could potentially be considered one, but still relatively minor. However, the execution is remarkable, and the results are impressive.

Overall, all reviewers recognize the significance of the paper. I'd urge the authors carefully consider the reviewers recommended revision, including ablation (reviewer vbjN) and reproducibility (reviewer c6Ub).